# H-MINT: MODELING POCKET-LIGAND BINDING WITH HIERARCHICAL MOLECULAR INTERACTION NETWORK

**Yanru Qu**[1*]    **Yijie Zhang**[3,4*]    **Wenjuan Tan**[5,6]    **Xiangzhe Kong**[5,6]    **Xiangxin Zhou**[7]

**Chaoran Cheng**[1,2]    **Mathieu Blanchette**[3,4]    **Jiaxuan You**[1]    **Ge Liu**[1,2†]

[1]Department of Computer Science, UIUC
[2]DOE Center for Advanced Bioenergy and Bioproducts Innovation, UIUC
[3]School of Computer Science, McGill University
[4]MILA-Québec AI Institute
[5]Department of Computer Science and Technology, Tsinghua University
[6]Institute for AI Industry Research (AIR), Tsinghua University
[7]School of Artificial Intelligence, University of Chinese Academy of Sciences
{yanruqu2, jiaxuan, geliu}@illinois.edu;
yj.zhang@mail.mcgill.ca

## ABSTRACT

Accurate molecular representations are critical for drug discovery, and a central challenge lies in capturing the chemical environment of molecular fragments, as key interactions, such as H-bond and $\pi$ stacking, occur only under specific local conditions. Most existing approaches represent molecules as atom-level graphs; however, atom-level representations can hardly express higher-order chemical context (*e.g.*, stereochemistry, lone pairs, conjugation). Fragment-based methods (*e.g.*, principal subgraph, predefined functional groups) fail to preserve essential information such as chirality, aromaticity, and ionic states. This work addresses these limitations from two aspects. (i) **OverlapBPE tokenization**[1]. We propose a novel data-driven molecule tokenization method. Unlike existing approaches, our method allows overlapping fragments, reflecting the inherently fuzzy boundaries of small-molecule substructures and, together with enriched chemical information at the token level, thereby preserving a more complete chemical context. (ii) **h-MINT model**. OverlapBPE induces many-to-many atom-fragment mappings, which necessitate a new hierarchical architecture. We therefore develop a hierarchical molecular interaction network capable of jointly modeling interactions at both atom and fragment levels. By supporting fragment overlaps, the model naturally accommodates the many-to-many atom-fragment mappings introduced by the OverlapBPE scheme. Extensive evaluation against state-of-the-art methods shows our method improves binding affinity prediction by 2-4% Pearson/Spearman correlation on PDBBind and LBA, enhances virtual screening by 1-3% in key metrics on DUD-E and LIT-PCBA, and achieves the best overall HTS performance on PubChem assays. Further analysis demonstrates that our method effectively captures interactive information while maintaining good generalization.

## 1 INTRODUCTION

Precise modeling of protein-ligand interactions is pivotal for fundamental tasks, such as binding affinity prediction and virtual screening, in early-stage drug discovery (R Laurie & Jackson, 2006). Beyond pharmaceutical applications, understanding enzyme pocket-ligand interactions is also critical for rational enzyme engineering, where small-molecule substrates or cofactors bind to catalytic

---

[*]Equal contribution.
[†]Corresponding author.
[1]BPE (Byte Pair Encoding) (Sennrich et al., 2016) is a statistical tokenization method that iteratively merges the most frequent pairs of symbols in a corpus to generate compact and efficient subword units.

pockets to regulate activity, stability, and specificity. Such capabilities are particularly important in bioenergy and bioproduct applications, where engineered enzymes are designed to efficiently convert biomass-derived substrates into sustainable fuels and chemicals. To accurately decipher these interactions, it is essential to build expressive representations to fully capture the chemical environment of the molecules. Most existing methods represent molecules as atom-level graphs (Zhou et al., 2023); however, it is questionable whether they are able to learn essential chemical information (*e.g.*, stereochemistry, lone pairs, conjugation) just from atomic tokens (Wigh et al., 2022). Another line of work employs molecular fragmentation (*e.g.*, Principal Subgraph (Kong et al., 2022b)) to preserve local contexts for atoms, in accordance with the intuition that many physicochemical properties occur at the fragment level (Murray & Rees, 2009). Nevertheless, these methods still fail to preserve crucial information like chirality, aromatic bond integrity, and ionic states, which stems from their naive partitioning of molecules into disjoint sets.

This work addresses the above limitations by integrating innovations in both molecular representations and model architectures. (i) We propose a novel and frequency-based molecular tokenization (*i.e.*, fragmentation) algorithm, **OverlapBPE**, which preserves essential chemical knowledge (*e.g.*, chirality, aromaticity, charges). Specifically, we enrich the atomic representation with their properties (*e.g.*, charges) and further incorporate 3D conformations during an iterative tokenizing process to extract frequently occurring fragments. To maintain the integrity of aromatic systems, we further enable atom overlaps between mined fragments, leading to a many-to-many mapping between atoms and fragments. For example, a Naphthalene (`c1c2ccccc2ccc1`) can be tokenized into 2 benzene rings (`c1ccccc1`) that share 2 aromatic C atoms. While the many-to-many mapping is necessary for aromatic integrity, it also poses an additional challenge on the model architecture, as most existing hierarchical molecular networks only support 1-1 mapping between atoms and fragments. (ii) OverlapBPE induces many-to-many atom-fragment mappings that existing hierarchical models are not designed to handle. To address this, we develop the hierarchical Molecular Interaction NeTwork (**h-MINT**), which explicitly supports overlapping atom-fragment structures. h-MINT introduces a bilevel attention mechanism that allows bidirectional information flow between atoms and overlapping fragments, and further expands fragment-level relations into atom-level geometric edges. This hierarchical yet equivariant design enables the model to capture multi-scale interaction patterns while maintaining global consistency.

Extensive experiments exhibit the superiority of our method over existing baselines, highlighting 2-4% performance gains on Pearson/Spearman correlation for binding affinity prediction, 1-3% gains on key metrics for virtual screening, and the best overall performance in HTS. Further analysis shows that our tokenization captures important inductive bias, making our model robust to noise and maintaining good generalization under different settings, indicating great potential for real-world applications[2].

## 2 BACKGROUND AND RELATED WORK

**Fragment-Based Molecular Tokenization.** Fragmentation partitions atom-level molecular graph into coarse units that capture meaningful substructural features. Early work relied on hand-crafted junction-tree rules or predefined fragment libraries (Jin et al., 2018; 2020; Yang et al., 2021). To reduce manual bias, subsequent studies adopted unsupervised frequent-subgraph mining to construct fragmentation rules in a data-driven manner: the underlying search is NP-hard (Kuramochi & Karypis, 2001; Jazayeri & Yang, 2021), but approximate algorithms make it tractable in practice (Inokuchi et al., 2000; Yan & Han, 2002; Nijssen & Kok, 2004; Geng et al., 2023). Byte-pair encoding (BPE), which iteratively merges the most frequent token pairs to build a compact vocabulary, has also been adapted to 2D molecular data (Li & Fourches, 2021; Ucak et al., 2023; Shen & Póczos, 2024). Our work is closely related to PS-VAE (Kong et al., 2022b), which proposes a data-driven tokenization that automatically mines and merges the most frequent, maximally-sized molecular fragments (*i.e.*, *principal subgraph*). However, essential chemical information, such as stereochemistry and conjugation, is usually neglected. In contrast, our proposed OverlapBPE tackles these challenges by enriching atomic properties, involving 3D stereochemistry, and enabling overlapping between fragments.

**Molecular Interaction Modeling.** Recent deep-learning frameworks for biomolecular modeling seek to couple fine-grained molecular representations interacting at multiple spatial resolutions.

---

[2]Code & checkpoints: https://github.com/Atomu2014/hmint

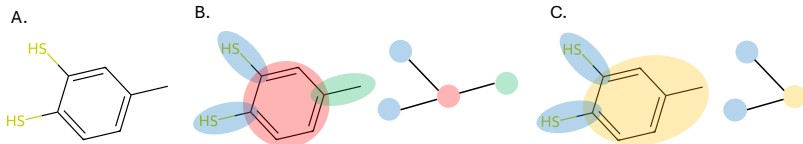

Figure 1: **Illustration of the OverlapBPE tokenization process**. (i) Starting from the molecule in A, we first extract all basic tokens from the atom graph. (ii) After identifying all basic tokens, we obtain the initial fragments (left) and token graph (right), as shown in B, which contains 4 tokens in 3 types: c1ccccc1 (freq=3778), Cc (freq=3496), and Sc (freq=637). (iii) We then enumerate all adjacent token pairs and identify the highest-frequency composite token from the final vocabulary $\Phi_{\text{final}}$, namely Cc1ccccc1 (freq=2458). (iv) Merging c1ccccc1 and Cc into a new token, we obtain new fragments and token graph, as shown in C, containing 3 tokens in 2 types: Cc1ccccc1 (freq=2458), Sc (freq=637). (v) Continue enumerating adjacent pairs in token graph C; no matched token found in vocabulary, the algorithm terminates.

At the finest scale, atom-level graphs are usually processed with $E(n)$-equivariant or directional message-passing networks, capturing local physics with high fidelity (Schütt et al., 2017; Satorras et al., 2021; Xu et al., 2022; Hoogeboom et al., 2022; Atz et al., 2021; Zaidi et al., 2022; Townshend et al., 2020). To reason over chemistry that spans several bonds such as aromatic conjugation, hydrogen-bond lattices and $\pi$-stacking, researchers introduce coarser views by pooling atoms into residues or surfaces for proteins (Jin et al., 2022; Anand & Achim, 2022; Somnath et al., 2021; Wang et al.), dual graphs for inverse folding (Gao et al., 2022), and functional-groups mined from small-molecule graphs (Jin et al., 2018; Kong et al., 2022b; Geng et al., 2023). Cross-molecule modules then embed ligand and receptor in a shared 3-D frame and predict poses or affinities with regression, contrastive, or diffusion objectives (Kong et al., 2022a; Luo et al., 2022; Stärk et al., 2022; Kong et al., 2024; Gao et al., 2023). Although these approaches broaden the receptive field, they often remain confined to a single resolution and are unable to enable bidirectional information flow between atoms and their corresponding substructures. Compared with prior hierarchical GNN approaches, our h-MINT introduces an atom-token overlap mechanism and expands token-level relations into atom-level geometric edges, enabling more flexible cross-scale information flow and yielding fine-grained yet globally consistent interaction modeling.

## 3 METHODOLOGY

### 3.1 OVERLAPBPE TOKENIZATION

We reveal that the failures of existing fragment-based methods stem from the naive partitioning of molecules into disjoint sets. To address these limitations, we propose a new tokenization approach that permits atom overlaps between mined fragments, enabling more expressive, coherent, and chemically meaningful molecular representations.

**Atom Graph.** An atom graph can be represented as a property graph $G^a = (V^a, E^a)$, where $V^a = \{a_i\}$ is a set of atoms, $E^a \subset V^a \times V^a$ is the set of bonds, and each atom/bond has an associated element/bond type. A tokenization step maps a subgraph of $G^a$ with certain atoms and bonds into a **fragment/token**[3] $(V', E') \rightarrow f \in \Phi$, where $(V', E') = G^a[V'] \subset 2^V \times 2^E$ is an induced subgraph, and $\Phi$ is the set of tokens.

**Token Graph.** A fragment/token graph $G^f$ is constructed from an atom graph $G^a$ through contracting subgraphs in $G^a$ as a single node while preserving connectivity. Each token thus represents an induced subgraph, and two tokens are connected if they share one or more atoms. This construction allows multiple tokens to overlap on shared atoms, forming a non-disjoint cover of the molecular graph. To avoid breaking important substructures and support token overlap, we first identify a set of **basic tokens** $\Phi_{\text{basic}}$, which includes smallest fragments that are chemically meaningful and should be preserved during partitioning. We include all single atoms, bonds, and rings collected from the

---

[3] We use the terms *token* and *fragment* interchangeably to refer to salient patterns mined from the atom graph.

training set in $\Phi_{\text{basic}}$[4] to guarantee the token set is complete, yet not all of the basic tokens will be added to the final vocabulary. In practice, we first replace all rings with tokens, then bonds, and lastly atoms to make sure all atoms and bonds are covered in the token graph $G^f$. In $G^f$, each node is a basic token, and the tokens are connected through sharing atoms (or disconnected when it is an ion).

$$G^f = (V^f, E^f), \quad \mathcal{T}^f : V^f \to \Phi_{\text{basic}} \tag{1}$$

where $V^f$ is the set of tokens, $E^f \subseteq V^f \times V^f$ encodes their adjacency (*e.g.*, two tokens share atoms or connected by bonds), and $\mathcal{T}^f$ maps each fragment to its token type in $\Phi_{\text{basic}}$. Besides, the following tokenization process follows a bottom-up BPE merge fashion[5], which also guarantees that the basic tokens will not be broken.

**Frequency-Based Vocabulary Setup.** After we get the token graph $G^f$ consisting of basic tokens, we iteratively discover new **composite tokens** and update the vocabulary $\Phi_{\text{comp}}$ as follows:

1. For a given token graph $G^f = (V^f, E^f)$, enumerate all adjacent token pairs $(f_i, f_j) \in E^f$ as composite token candidates: $\mathcal{C} = \{\text{Merge}(f_i, f_j) \mid (f_i, f_j) \in E^f\}$ where $\text{Merge}(f_i, f_j)$ denotes the operation of combining two neighboring tokens into a larger token. Token frequency is computed over the training corpus.

2. Select the most frequent token $f^* \in \mathcal{C}$ and add it to the vocabulary: $\Phi_{\text{comp}} \leftarrow \Phi_{\text{comp}} \cup \{f^*\}$.

3. Update all token graphs $\{G^f\}$ in the corpus by replacing each occurrence of $f^*$ with a new hyper node. Note that the original tokens in $V^f$ will not be removed from $G^f$ unless all of its adjacent composite candidates have been merged.

4. Repeat steps 1-3 until a stopping criterion (*e.g.*, iteration steps) is met.

5. Filter $\Phi_{\text{basic}}$ and $\Phi_{\text{comp}}$ with minimum frequency threshold to make sure a proper vocabulary size $\Phi_{\text{final}} = \{f \in \Phi_{\text{basic}} \cup \Phi_{\text{comp}} \mid \text{freq}(f) > t\}$.

**OverlapBPE Tokenization.** After obtaining the vocabulary, we can tokenize an atom-bond graph $G^a$ following the frequency of tokens in $\Phi_{\text{final}}$. Figure 1 illustrates the tokenization process. For more details, please refer to Appendix C.1.

1. Find all basic tokens $\Phi_{\text{basic}}$ from $G^a$ and convert $G^a$ into token graph $G^f$. For basic tokens in $G^f$, we preserve their token identifier if it's in $\Phi_{\text{final}}$, otherwise, we replace it with a special identifier (*e.g.*, <ring>) to make sure not breaking it in the following.

2. For $G^f$, enumerate all adjacent token pairs $(f_i, f_j) \in E^f$ and find the matching token $f^* \in \Phi_{\text{final}}$ with the highest frequency that can be merged from pair(s) $(f_i, f_j) \in E^f$. Replace pair(s) with $f^*$ and update $G^f$. Note that a token $f_i \in V^f$ will not be removed from $G^f$ unless $f_i$ and all its adjacent pairs $(f_i, f_j) \in E^f$ have been merged.

3. Repeat step 2 until no token pairs match tokens in $\Phi_{\text{final}}$.

**Chemical Information Incorporation.** Our vocabulary can be easily extended to incorporate domain knowledge. (i) To distinguish **Chirality**, the algorithm operates on 2D molecular graphs augmented with 3D conformer information. The use of 3D coordinates ensures that stereochemistry is preserved during tokenization. As a result, each token is assigned a unique isomeric SMILES string as its vocabulary identifier, encoding both atomic connectivity and chirality. For example, L-lactic and R-lactic are represented by `C[C@H](O)C(=O)O` and `C[C@@H](O)C(=O)O`, respectively. (ii) **Aromatic integrity** is ensured by two complementary mechanisms. First, aromatic rings are treated as indivisible basic tokens, which are preserved throughout the bottom-up merge process. Second, the overlapping tokenization strategy allows for progressive merging of neighboring tokens, enabling the discovery of extended conjugated systems while maintaining aromatic consistency. (iii) To properly encode **atomic attributes**, such as charged and aromatic atoms, we assign explicit identifiers to atoms with formal charges and/or aromatic participation. For instance, `[Cl-]` denotes a negatively

---

[4]Aromatic rings are treated as single delocalized $\pi$ systems to preserve conjugation.

[5]In BPE, tokenization starts from an initial vocabulary of basic tokens (*e.g.*, characters or word pieces), and iteratively merges the most frequent adjacent pairs. Since merging only combines existing tokens without splitting them, the original basic tokens remain intact.

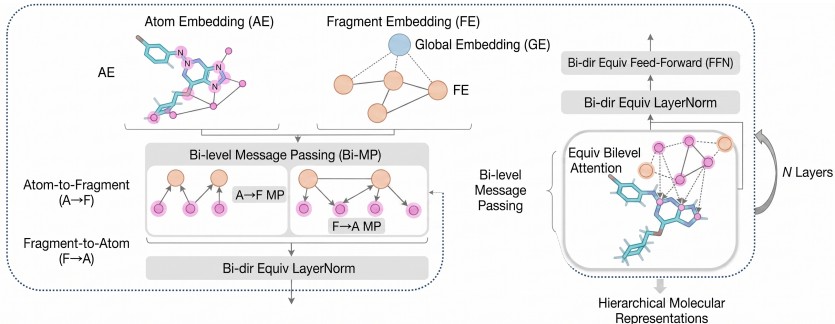

Figure 2: **Overall model architecture**. (A) Global node, fragments, and atoms in the ligand molecule of an input pair. (B) The aggregation of fragment and global embeddings. (C) An encoder layer of h-MINT. *Note*: Solid lines indicate connection within the same level. Dashed lines indicate connections across different levels.

charged chlorine atom, while `[n+]` indicates a positively charged aromatic nitrogen. In contrast to standard SMILES, which often omit such details when representing isolated atoms, our token identifiers explicitly preserve these chemically significant properties.

### 3.2 H-MINT

In this section, we introduce the hierarchical Molecular Interaction NeTwork (h-MINT), designed to accommodate the overlapping atom-fragment mappings induced by OverlapBPE. Existing hierarchical molecular networks typically rely on disjoint atom-to-fragment assignments, limiting their ability to model many-to-many relationships and bidirectional cross-scale interactions. To overcome this restriction, we develop an SE(3)-equivariant graph Transformer that (i) adapts its self-attention, feed-forward, and normalization layers to maintain equivariance, and (ii) explicitly preserves atom-token overlap to enable flexible cross-scale information flow. The overall architecture is in Figure 2.

#### 3.2.1 MODEL INPUT AND EMBEDDING

The model receives a pocket-ligand pair as input, including their atoms $(V_p^a, V_l^a)$[6], tokens $(V_p^f, V_l^f)$, and atom-token mapping $\mathcal{T}_p \subseteq V_p^a \times V_p^f, \mathcal{T}_l \subseteq V_l^a \times V_l^f$. Note that: (i) each node in $V^a$ or $V^f$ has a type (*e.g.*, element type `N`, token type `[Cl-]`). (ii) The mapping function $\mathcal{T}$ has 2 parts to map an atom index to its associated token indices $\mathcal{T}_{a2f}$ and vice versa $\mathcal{T}_{f2a}$, yet do not operate on atom and token types. Thus, the embeddings can be obtained:

$$H_p^0 = \text{Embed}(V_p^a, V_p^f, \mathcal{T}_p), \quad H_l^0 = \text{Embed}(V_l^a, V_l^f, \mathcal{T}_l), \tag{2}$$

$$H^0 = \text{Embed}(V^a) + \text{ScatterMean}(\text{Embed}(V^f), \mathcal{T}_{f2a}) + \text{Embed}(\text{Pos}(V^a)), \tag{3}$$

where we use `scatter_mean` to aggregate an atom's associated token types information, and $\text{Pos}(\cdot)$ simply maps an atom to its position code.[7] To collect global information from atoms and tokens, we also augment special `<global>` tokens to all node lists $V_p^a, V_l^a, V_p^f, V_l^f$. Some example inputs can be found in the Appendix C.1.

#### 3.2.2 BUILDING HIERARCHICAL GRAPH

The embedding layer only encodes type and position information, and also uses $\mathcal{T}_p, \mathcal{T}_l$ to map token type embedding into atom embedding dimensions. In this section, we further consider 3D structures and build hierarchical graphs for message passing, as shown in Figure 2 B. Before we start building the graph, we need to specify that the edges are directed, which means for any token-level edge $(f_i, f_j)$ or atom-level edge $(a_i, a_j)$, the former node will receive a message from the latter.

---

[6]We use subscript $_p$ and $_l$ for pocket and ligand, superscript $^a$ and $^f$ to distinguish atom and token (fragment).

[7]The position code for a residue atom is its atom name, *e.g.*, `CA` $\rightarrow$ `A`, for a molecular atom is a special token `<sm>`. We leave the complete position code table in the Appendix C.1.

**KNN Token Graph.** In the token level, every pair of input contains a sequence of pocket tokens and a sequence of ligand tokens, plus 2 augmented global tokens, as follows:

$$[V_p^f; V_l^f] = [f_{p,g}, f_{p,0}, f_{p,1}, f_{p,2}, \ldots, f_{l,g}, f_{l,0}, f_{l,1}, f_{l,2}, \ldots], \tag{4}$$

where we use $f_p$ and $f_l$ to represent pocket and ligand tokens, and subscript $f_{,g}$ and $f_{,i}$ to denote corresponding global token and $i$-th token. In this step, we construct a KNN graph for non-global tokens, where the distance between token $f_i$ and $f_j$ is the minimum distance between all their atoms,

$$\text{dist}(f_i, f_j) = \min_{a_s \in f_i, a_t \in f_j} \text{dist}(a_s, a_t), \tag{5}$$

For the two global tokens, we connect them with their following tokens to aggregate information within each pocket and ligand, and we connect the 2 global tokens to exchange pair information. Therefore we get the set of edges $E_{\text{KNN}}^f = \{(f_i, f_j) | f_i \in V_p^f \cup V_l^f, f_j \in \text{KNN}(f_i)\} \cup \{(f_{,g}, f_{,i}) | f_{,i} \in V^f\} \cup \{(f_{p,g}, f_{l,g})\}$. For simplicity, we use $\text{KNN}(f_i) = \{f_j | (f_i, f_j) \in E_{\text{KNN}}^f\}$ to denote neighborhood of $f_i$.

**Token-expanded Atom Graph.** The atom graph is constructed through expanding each token-level edge from $E_{\text{KNN}}^f$ into several atom-level edges. More specifically, for a token-level edge in $(f_i, f_j) \in E_{\text{KNN}}^f$, we connect each atom with atoms from the other token. In practice, to control the number of atom-level edges, for an atom $a_i$, we connect $a_s \in f_i$ with the k-nearest atoms in $f_j$. Through this expansion, we obtain a set of atom-level edges that contain short-range interactions within atoms' neighborhoods, as well as long-range interactions bridged by token-level edges. Finally, we get the set of atom-level edges $E_{\text{knn}}^a = \{(a_s, a_t) | a_s \in f_i, a_t \in \text{knn}(f_j, a_s), (f_i, f_j) \in E_{\text{KNN}}^f\}$, and we use $\text{knn}(f_i, f_j)$ to denote the set of atom-level edges expanded from $(f_i, f_j)$.

### 3.2.3 Bilevel Message Passing

Given the aforementioned atom embeddings $[H_p^0; H_l^0]$ and token-grouped atom-level edges $E_{\text{knn}}^a$, we now introduce our bilevel message passing through equivariant bilevel graph attention.

**Equivariant Bilevel Graph Attention.** For a token edge $(f_i, f_j) \in E_{\text{KNN}}^f$ and their expanded atom edges $\{(a_s, a_t) | a_s \in f_i, a_t \in \text{knn}(f_j, a_s)\}$, we first compute *atom-level cross attention* based on input atom embeddings $H^{l-1}$, where $l$ is the current layer:

$$[\mathbf{Q}^l; \mathbf{K}^l; \mathbf{V}^l] = \text{Linear}(H^{l-1}), \tag{6}$$

$$\text{Score}: S_{i,j}[a_s, a_t] = \text{MLP}(\mathbf{Q}^l[a_s], \mathbf{K}^l[a_t], \text{RBF}(D[a_s, a_t]), \mathbf{e}_{i,j}), \tag{7}$$

$$\text{Attention Weight}: \alpha_{i,j}[a_s, a_t] = \text{Softmax}_{a_t \in \text{knn}(f_j, a_s)}(S_{i,j}[a_s, a_t]W_\alpha), \tag{8}$$

where $\mathbf{Q}^l[a_s]$, $\mathbf{K}^l[a_t]$ are query and key vectors of $a_s, a_t$, $\text{RBF}(D[a_s, a_t])$ embeds the relative position of $a_t$ to $a_s$, $\mathbf{e}_{i,j}$ is the edge type embedding of $(f_i, f_j)$[8], and $W_\alpha$ is to project the scores into scalars. Through Softmax in Eq. 8, $a_s$ is able to aggregate messages from $a_t \in \text{knn}(f_j, a_s)$.

The *token-level cross attention* is defined through aggregating all atom-level edges expanded from the token-level edge.

$$\text{Score}: S_{i,j} = \frac{1}{|\text{knn}(f_i, f_j)|} \sum_{(a_s, a_t) \in \text{knn}(f_i, f_j)} M_{i,j}[a_s, a_t], \tag{9}$$

$$\text{Attention Weight}: \beta_{i,j} = \text{Softmax}_{f_j \in \text{KNN}(f_i)}(S_{i,j}W_\beta), \tag{10}$$

where $W_\beta$ projects the scores into scalars. Basically, $S_{i,j}$ aggregates all atom-level edges expanded from $(f_i, f_j)$. Then we have the following message passing and embedding update:

$$\mathbf{m}_{i,j}[a_s] = \sum_{a_t \in \text{knn}(f_j, a_s)} \alpha_{i,j}[a_s, a_t]\mathbf{V}^l[a_t], \tag{11}$$

$$\mathbf{m}_i[a_s] = \sum_{f_j \in \text{KNN}(f_i)} \beta_{i,j}\text{MLP}(\mathbf{m}_{i,j}[a_s]), \tag{12}$$

$$H^l[a_s] \leftarrow H^{l-1}[a_s] + \text{ScatterMean}(\mathbf{m}_i[a_s], \mathcal{T}_{f2a}). \tag{13}$$

---

[8]We use different edge types to distinguish intra- and inter-molecule edges.

Table 1: **Mean and standard deviation of three runs on the PDBBind benchmark**. The best results are marked in **bold**, and the second best are underlined. Baseline results are taken from (Wang et al.; Kong et al., 2024). Details in Appendix D.

| | Model | RMSE ↓ | Pearson ↑ | Spearman ↑ |
|---|---|---|---|---|
| Separate Encoder | DeepDTA | 1.866 ± 0.080 | 0.472 ± 0.022 | 0.471 ± 0.024 |
| | Bepler and Berger's | 1.985 ± 0.006 | 0.165 ± 0.006 | 0.152 ± 0.024 |
| | TAPE | 1.890 ± 0.035 | 0.338 ± 0.044 | 0.286 ± 0.124 |
| | ProtTrans | 1.544 ± 0.015 | 0.438 ± 0.053 | 0.434 ± 0.058 |
| | MaSIF | 1.484 ± 0.018 | 0.467 ± 0.020 | 0.455 ± 0.014 |
| | IEConv | 1.554 ± 0.016 | 0.414 ± 0.053 | 0.428 ± 0.032 |
| | Holoprot-Full Surface | 1.464 ± 0.006 | 0.509 ± 0.002 | 0.500 ± 0.005 |
| | Holoprot-Superpixel | 1.491 ± 0.004 | 0.491 ± 0.014 | 0.482 ± 0.032 |
| | ProNet-Amino Acid | 1.455 ± 0.009 | 0.536 ± 0.012 | 0.526 ± 0.012 |
| | ProNet-Backbone | 1.458 ± 0.003 | 0.546 ± 0.007 | 0.550 ± 0.008 |
| | ProNet-All-Atom | 1.463 ± 0.001 | 0.551 ± 0.005 | 0.551 ± 0.008 |
| | ESM-2 + fingerprint | 1.537 ± 0.001 | 0.455 ± 0.013 | 0.433 ± 0.009 |
| Joint Encoder | GVP | 1.594 ± 0.073 | - | - |
| | Atom3D-3DCNN | 1.416 ± 0.021 | 0.550 ± 0.021 | 0.553 ± 0.009 |
| | Atom3D-ENN | 1.568 ± 0.012 | 0.389 ± 0.024 | 0.408 ± 0.021 |
| | Atom3D-GNN | 1.601 ± 0.048 | 0.545 ± 0.027 | 0.533 ± 0.033 |
| | GET | 1.430 ± 0.007 | 0.586 ± 0.001 | 0.575 ± 0.002 |
| | GET-Murcko | 1.415 ± 0.010 | 0.590 ± 0.002 | 0.578 ± 0.003 |
| | GET-BRICS | 1.410 ± 0.008 | 0.592 ± 0.003 | 0.579 ± 0.004 |
| | GET-PS | 1.387 ± 0.015 | 0.601 ± 0.002 | 0.582 ± 0.005 |
| | Ours | **1.295 ± 0.001** | **0.640 ± 0.002** | **0.625 ± 0.002** |

where $\mathbf{V}^l[a_t]$ is the value vector of $a_t$ at layer-$l$. Stacking equivariant bilevel attention, equivariant feed-forward layer, and equivariant layer normalization together, we obtain an equivariant graph Transformer layer, as shown in Figure 2 C. For more details, please refer to Appendix C.2.

## 4 EXPERIMENTS AND RESULTS

In this section, we evaluate our OverlapBPE and h-MINT model in two fundamental drug discovery tasks: binding affinity prediction (Section 4.1) and virtual screening (Section 4.2). We conduct further experiments and case study for OverlapBPE in incorporating chemical information and representing chirality in Section 4.3.

### 4.1 BINDING AFFINITY PREDICTION

**Task Definition.** Given the 3D structure of a pocket-ligand pair, the task is to predict the binding affinity, *i.e.*, change in free energy upon binding. The input is a complex $(p, l)$ with 3D structure, and the output is a real value $y$. Performance is assessed with regression metrics such as root-mean-square error (RMSE) and Pearson/Spearman correlation against experimental affinities.

**Setup.** We follow Somnath et al. (2021); Wang et al. to conduct experiments on the well-established **PDBBind** benchmark (Wang et al., 2005) and split the 4,709 complexes according to sequence identity of the protein using a 30% threshold. We also employ the **LBA** dataset with its predefined splits from the Atom3D benchmark (Townshend et al., 2020). Both PDBBind and LBA comprise 3,507/466/490 protein-ligand complexes for training/validation/testing, with major difference in pre-processing. For the baseline models, we compare against a variety of approaches (Öztürk et al., 2018; Bepler & Berger, 2019; Rao et al., 2019; Elnaggar et al., 2022; Gainza et al., 2020; Hermosilla et al., 2020; Somnath et al., 2021; Wang et al.; Jing et al., 2021; Townshend et al., 2020; Schütt et al., 2017; Gasteiger et al., 2021; Zhou et al., 2023; Gao et al.; Feng et al.). We also include another baseline (ESM-2+fingerprint) which incorporates ESM-2 embedding to represent pockets and traditional fingerprints (Morgan+ERP+Avalon) to represent molecules. Details are provided in Appendix D. Among these models, we primarily focus our comparison with GET (Kong et al., 2024) and its variants, as it currently achieves the best performance and shares similarity with our method. Specifically, we

Table 2: **Mean and standard deviation of three runs on LBA prediction**. The baseline results are from Kong et al. (2024). Models with * are large pretrained models, results from (Gao et al.; Feng et al.). The best results are marked in **bold** and the second best are underlined. *Note:* To save space, we only report the best baseline settings from atom-level, fragment-level, and bi-level. Details in Appendix D.

| Best Repr. Setting | Model | RMSE ↓ | Pearson ↑ | Spearman ↑ |
|---|---|---|---|---|
| Atom-level | SchNet | $1.357 \pm 0.017$ | $0.598 \pm 0.011$ | $0.592 \pm 0.015$ |
| | EGNN | $1.358 \pm 0.000$ | $0.599 \pm 0.002$ | $0.587 \pm 0.004$ |
| | LEFTNet | $1.343 \pm 0.004$ | $0.610 \pm 0.004$ | $0.598 \pm 0.003$ |
| | ET | $1.367 \pm 0.037$ | $0.599 \pm 0.017$ | $0.584 \pm 0.025$ |
| | UniMol* | $1.434$ | $0.565$ | $0.540$ |
| | BigBind* | $1.340$ | $0.632$ | $0.620$ |
| | ProFSA* | $1.377$ | $0.628$ | $0.620$ |
| Frag-level | GemNet | $1.393 \pm 0.036$ | $0.569 \pm 0.027$ | $0.553 \pm 0.026$ |
| | Equiformer | $1.350 \pm 0.019$ | $0.604 \pm 0.013$ | $0.591 \pm 0.012$ |
| | DimeNet++ | $1.388 \pm 0.010$ | $0.582 \pm 0.009$ | $0.574 \pm 0.007$ |
| Bi-level | MACE | $1.372 \pm 0.021$ | $0.612 \pm 0.010$ | $0.592 \pm 0.010$ |
| | GET | $1.331 \pm 0.008$ | $0.618 \pm 0.005$ | $0.607 \pm 0.005$ |
| | GET-PS | $\underline{1.312 \pm 0.016}$ | $\underline{0.631 \pm 0.011}$ | $\underline{0.642 \pm 0.011}$ |
| | Ours | $\mathbf{1.276 \pm 0.011}$ | $\mathbf{0.660 \pm 0.001}$ | $\mathbf{0.661 \pm 0.001}$ |

include *GET-PS*, *GET-Murcko*, and *GET-BRICS*, which are GET models incorporating the Principal Subgraph tokenization (Kong et al., 2022b), Bemis-Murcko scaffolding (Bemis & Murcko, 1996), and BRICS tokenization (Wegscheid-Gerlach et al.).

**Results.** We employ OverlapBPE on the ligand molecules while utilizing residues as tokens for pockets, and conduct evaluations on PDBbind and LBA datasets separately. Table 1 and 2 report the mean and the standard deviation of the metrics across 3 runs for the PDBbind and LBA datasets. Our model demonstrates significantly better performance over baseline methods in the binding affinity prediction task. Our improvement suggests that our tokenization and modeling may better preserve interaction-relevant chemical features. Significance test and more detailed analysis are in Appendix D.

## 4.2 STRUCTURE-BASED VIRTUAL SCREENING

**Task Definition.** Given the 3D structure of a pocket $p$ and a library of ligands $\{l_i\}$, the task is to retrieve candidate ligands that are possible to bind with $p$. A Virtual Screening (VS) model learns a score $s_i$ for each pair $(p, l_i)$, and ranks all candidate ligands in descending order of $s_i$. The evaluation metrics include the area under the ROC curve (AUC), enrichment factor at a given top-$k$ threshold (EF@$k$), and BEDROC, which emphasize early-recognition of true binders.

**Setup.** Evaluation was performed on DUD-E (Mysinger et al., 2012) following preprocessing by Gao et al. (2023) and LIT-PCBA (Tran-Nguyen et al., 2020). DUD-E contains 22,886 active compounds with binding affinities across 102 protein targets (about 224 ligands per target), and for each active ligand provides around 50 decoys that match physicochemical properties but differ in 2D topology. The dataset is mainly used to test how well docking methods can distinguish true binders from non-binders. The LIT-PCBA dataset (Tran-Nguyen et al., 2020) is constructed from dose-response PubChem bioassays to mitigate the target and decoy selection biases found in other benchmarks. It comprises 15 protein targets with 7844 experimentally confirmed actives and 407381 inactive, which reflects realistic hit rates in high-throughput screening. We include other two benchmarks DEKOIS (Vogel et al., 2011), and JACS/Merck (Wang et al., 2015; Schindler et al., 2020) as used in (Feng et al., 2025) in Appendix E.4. We benchmark against classical docking tools (Halgren et al., 2004; Trott et al., 2009), early ML scoring functions (Durrant & Mccammon, 2011; Ballester et al., 2010; Stepniewska-Dziubinska et al., 2017; Zheng et al., 2019; Zhang et al., 2023), and contrastive learning frameworks DrugCLIP (Gao et al., 2023) and LigUnity (Feng et al., 2025).
**h-MINT adapter**: We observe that baseline models commonly benefit from pretrained UniMol model to improve representation quality. We therefore incorporate h-MINT as a lightweight adapter on top of the UniMol encoder to assess its effectiveness in conjunction with large pretrained models.
**PDBBind-only finetuning**: Since DrugCLIP and LigUnity use different data augmentations, for

Table 3: **Zero-shot Virtual Screening on DUD-E**. Baseline results are from (Gao et al., 2023; Feng et al., 2025). Models with * are trained with PDBBind data only for fair comparison. Details in Appendix E.

| | AUC (%) ↑ | BEDROC (%) ↑ | 0.5% | EF ↑ 1% | 5% |
|---|---|---|---|---|---|
| Glide-SP | 76.70 | 40.70 | 19.39 | 16.18 | 7.23 |
| Vina | 71.60 | - | 9.13 | 7.32 | 4.44 |
| NN-score | 68.30 | 12.20 | 4.16 | 4.02 | 3.12 |
| RFscore | 65.21 | 12.41 | 4.90 | 4.52 | 2.98 |
| Pafnucy | 63.11 | 16.50 | 4.24 | 3.86 | 3.76 |
| OnionNet | 59.71 | 8.62 | 2.84 | 2.84 | 2.20 |
| Planet | 71.60 | - | 10.23 | 8.83 | 5.40 |
| DrugCLIP * | 81.39 | 45.96 | 34.27 | 29.01 | 10.18 |
| LigUnity * | 81.69 | 46.01 | 34.44 | 29.07 | 10.26 |
| Ours * | **84.45** | **47.64** | **35.06** | **29.91** | **10.76** |

Table 4: **Zero-shot Virtual Screening on LIT-PCBA**. Baseline results are from Gao et al. (2023); Feng et al. (2025); Jia et al. (2024). Models with * are trained with PDBBind data only. Bold numbers indicate the best performance.

| | AUC (%) ↑ | BEDROC (%) ↑ | 0.5% | EF ↑ 1% | 5% |
|---|---|---|---|---|---|
| Surflex | 51.47 | - | - | 2.50 | - |
| Glide-SP | 53.15 | 4.00 | 3.17 | 3.41 | 2.01 |
| Planet | 57.31 | - | 4.64 | 3.87 | **2.43** |
| Gnina | 60.93 | 5.40 | - | 4.63 | - |
| DeepDTA | 56.27 | 2.53 | - | 1.47 | - |
| BigBind | 60.80 | - | - | 3.82 | - |
| DrugCLIP * | 58.15 | 4.12 | 4.11 | 3.08 | 2.27 |
| LigUnity * | 57.61 | 4.34 | 4.06 | 3.03 | 2.25 |
| Ours * | **57.77** | **6.27** | **7.01** | **5.20** | 2.18 |

a fair comparison, we train these models and our model with PDBBind (Wang et al., 2005) only, which comprises 16,744 pocket-ligand pairs, with any overlap with the test sets removed. Following the conventions (Feng et al., 2025; Gao et al., 2023), these models are initialized with pretrained checkpoint of UniMol (Zhou et al., 2023).

For detailed implementation, please refer to Appendix E.

**Results.** We report AUC, BEDROC, and enrichment factors (EF) on DUD-E and LIT-PCBA in the zero-shot setting in Table 3 and Table 4. Our model demonstrates strong generalization under the zero-shot setting, highlighting the expressive power of OverlapBPE and the effectiveness of the hierarchical design in h-MINT. Notably, our method consistently outperforms state-of-the-art baselines across all evaluation metrics, underscoring its capability to accurately capture fine-grained protein-ligand interaction patterns.

### 4.3 FURTHER EXPERIMENTS OF OVERLAPBPE

In this section, we provide additional analysis about our tokenization method's advantages in two aspects: incorporating chemical information and representing chirality.

**Incorporating chemical information.** Figure 3 illustrates a case study from the LBA dataset. Our tokenization preserves the integrity of the benzene ring and retains the positive charge of [N+], which is necessary for forming the pi-cation interaction between the ligand and the protein pocket. In contrast, the PS tokenization treats [N] as neutral and thus cannot capture this interaction. This difference contributes to more accurate affinity prediction, with our method achieving an error of 0.56 compared to 0.67 for the PS tokenization.

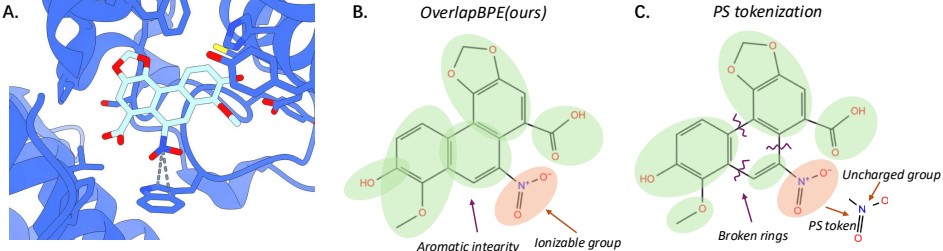

Figure 3: **OverlapBPE (ours) better preserves aromatic bond integrity, and ionic states**. (A) An interaction formed between the ligand and the protein pocket. The ligand contains a positively charged [N+], which forms two $\pi$-cation interactions with two aromatic rings in the protein pocket. (B) Representation using our tokenization method. Green colors indicate fragments without charge. Red colors indicate charged fragments. (C) Representation using the PS tokenization method, where green and red indicate fragments as above, but the [N+] charge is not preserved, and some rings are broken.

**Representing chirality.** To further validate the effectiveness of our overlap tokenization and its capability to represent chirality, we followed MolKGNN (Liu et al., 2023) and conducted high-throughput screening (HTS) experiments on PubChem assays as a binary classification task. Specifically, we only use the OverlapBPE tokenization with XGBoost (Chen & Guestrin, 2016) and do not adopt the h-MINT model for two reasons: (i) the dataset lacks pocket structures, and (ii) HTS places stringent demands on efficiency. We evaluate OverlapBPE on 8 PubChem HTS assays in Table 5. The baselines include strong atom-level GNN models such as SchNet (Schütt et al., 2017), SphereNet (Liu et al., 2022), ChiRo (Adams et al., 2021), KerGNNs (Feng et al., 2022) and MolKGNN (Liu et al., 2023). For OverlapBPE, we investigate two variants: a *chiral* vocabulary that preserves chirality in tokens, and a *non-chiral* vocabulary that omits stereochemical information. All baselines rely on extensive molecular features and complex model architectures, whereas our method only utilizes tokenized bag-of-words features combined with XGBoost for classification. We report early-enrichment performance using $\log\text{AUC}_{[0.001,0.1]}$ in Table 5 and can observe that: (i) Our chiral method significantly outperforms non-chiral one, showing the importance of chirality information in HTS tasks; (ii) Our method outperforms all baselines in average ranking, and even exceeds ChiRo and MolKGNN, which are designed to represent molecular chirality; (iii) Leveraging XGBoost's lightweight feature, our method completes training and prediction within 1 second.

Table 5: **Early-enrichment performance on PubChem HTS assays**. Metric is $\log\text{AUC}_{[0.001,0.1]}$ (higher is better). **Ours (chiral)** preserves the chirality of tokens and **Ours (non-chiral)** does not. Bold numbers indicate the best method per dataset. Baseline results are taken from Liu et al. (2023).

| PubChem AID | MolKGNN | SchNet | SphereNet | DimeNet++ | ChiRo | KerGNN | Ours (chiral) | Ours (non-chiral) |
|---|---|---|---|---|---|---|---|---|
| 435008 | **0.255** | 0.187 | 0.215 | 0.203 | 0.168 | 0.147 | 0.221 | 0.211 |
| 1798 | 0.174 | 0.195 | 0.196 | 0.208 | 0.165 | 0.078 | 0.217 | **0.282** |
| 435034 | 0.227 | 0.246 | 0.230 | 0.235 | 0.211 | 0.179 | **0.281** | 0.261 |
| 2258 | 0.301 | 0.240 | **0.380** | 0.340 | 0.251 | 0.195 | 0.265 | 0.246 |
| 463087 | 0.390 | 0.332 | **0.399** | 0.389 | 0.258 | 0.150 | 0.338 | 0.322 |
| 488997 | 0.303 | 0.319 | 0.309 | 0.315 | 0.193 | 0.081 | **0.384** | 0.376 |
| 2689 | **0.415** | 0.324 | 0.401 | 0.367 | 0.351 | 0.264 | 0.348 | 0.343 |
| 485290 | **0.498** | 0.333 | 0.450 | 0.463 | 0.295 | 0.223 | 0.474 | 0.341 |
| Avg. Rank | 3.250 | 5.250 | 3.125 | 3.375 | 6.375 | 8.000 | **2.625** | 4.000 |

## 5 DISCUSSION AND FUTURE WORK

We introduced OverlapBPE and h-MINT, an efficient tokenization-plus-learning framework for protein-ligand interactions that achieves superior performance on affinity prediction and virtual screening. Our results demonstrate that explicitly modeling fragment overlaps is not only feasible but necessary for preserving chemically meaningful context, leading to consistent performance gains across diverse tasks. For future work, we plan to validate OverlapBPE on molecular tasks and extend h-MINT to docking pose prediction and broader drug design or enzyme engineering tasks.

ACKNOWLEDGMENTS

Research was supported in part by the Molecule Maker Lab Institute: An AI Research Institutes program supported by NSF under Award No. 2505932, and the DOE Center for Advanced Bioenergy and Bioproducts Innovation (U.S. Department of Energy, Office of Science, Biological and Environmental Research Program under Award Number DE-SC0018420). Any opinions, findings, and conclusions or recommendations expressed in this publication are those of the author(s) and do not necessarily reflect the views of the U.S. Department of Energy. Yijie Zhang received support from the Canada First Research Excellence Fund and the Fonds de recherche du Québec awarded to the D2R Initiative at McGill University.

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

## A   ETHICS STATEMENT

Small-molecule modeling plays a critical role in drug discovery, with broad potential applications in therapeutic development, virtual screening, and rational design of ligands targeting protein pockets. Advances in representation learning and interaction modeling offer new opportunities to accelerate discovery and improve our understanding of molecular interactions, which may positively impact medicine, biotechnology, and related fields.

At the same time, we recognize that such computational approaches also carry potential risks, particularly regarding misuse in unsafe or unethical drug design. To mitigate these risks, this study is conducted exclusively on publicly available datasets and strictly follows established ethical guidelines. We advocate for the responsible research and application of molecular modeling methods to ensure their development contributes to societal benefit.

## B   REPRODUCIBILITY STATEMENT

We ensure that the training data, training and inference procedures, and result evaluations are all reproducible. The appendix provides all necessary details and offers a comprehensive explanation of each component of this work. The datasets used are publicly available, and the model implementation is based on the open-source GET (Kong et al., 2024), LigUnity (Feng et al., 2025) and MolKGNN (Liu et al., 2023) codebases. The code and models used for evaluation are also publicly accessible and cited in the appendix. Furthermore, we describe the training hyperparameters in detail in the appendix, thereby ensuring that the entire experimental process is fully reproducible.

## C   METHOD

### C.1   OVERLAPBPE

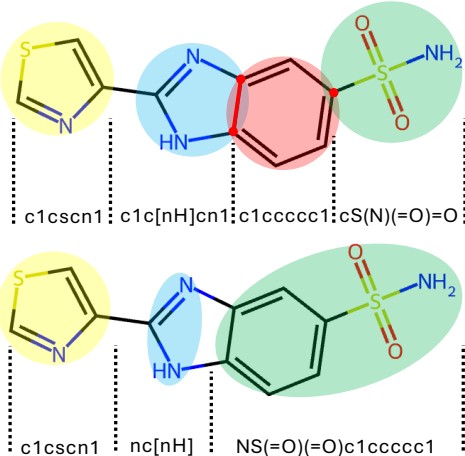

Figure 4: **Comparison of overlap (top) and non-overlap tokenization (bottom).**

**Overlap vs. Non-overlap Tokenization**    In Figure 4, we show an example molecule tokenized in 2 different ways. We use different colors to highlight the tokenized fragments. From the top figure, we can see the molecule has 4 tokens, with 3 of the tokens sharing atoms with other tokens (we also highlight the shared atoms and bonds in red). From the bottom figure, we can see that an aromatic bond is broken and forms 2 disjoint tokens. From this figure, we can identify the difference between overlap-tokenization and non-overlap-tokenization. With overlap-tokenization better preserving the local chemical environment and better respecting the fuzzy boundaries of substructures in small molecules, we believe our novel OverlapBPE is potent to boost small molecular representation, interaction, and even generation tasks.

**Example Input.** Here we show some example inputs to the encoder, more specifically, the embedding layer. The embedding layer takes 3 inputs: atom sequences, atom-level position codes, and token sequences, for pairs of pocket-ligands.

$$[V_p^a; V_l^a]: [\texttt{<g\_atom>, N, C, C, O, C, ..., <g\_atom>, C, C, ...}] \tag{14}$$

$$[\text{Pos}(V_p^a); \text{Pos}(V_l^a)]: [\texttt{<g\_pos>, `', A, `', `', B, ..., <g\_pos>, sm, sm, ...}] \tag{15}$$

$$[V_p^f; V_l^f]: [\texttt{<g\_frag>, ALA, ..., <g\_frag>, c1cscn1, ...}] \tag{16}$$

**Position Code.** Certain types of molecules have conventional position codes to distinguish different atoms with the same element type in the same fragment, *e.g.*, CA and CB in residues. For small molecules, since there are no such conventional position codes, we simply use sm as the position code for all atoms in small molecules. In addition, we also add some special position codes for special tokens, e.g., <global>, <mask>, <pad>.

**Tokenization Overhead.** Compared to the PS tokenizer and GET, the computational overhead of OverlapBPE and h-MINT mainly stems from the repeated calculation of atoms. We performed tokenization using OverlapBPE and PS on the 3,507 molecules in the LBA training set, and the results are shown in Table 6.

Table 6: Comparison of tokenization statistics.

| Tokenizer | Avg # tokens / mol | Avg # atoms / token |
|---|---|---|
| OverlapBPE | 7.95 | 4.5 |
| Principal Subgraph | 8 | 3.39 |

For tokenization, the number of atoms processed by OverlapBPE is 1.32 times that of PS: $(7.95 \times 4.5)$ / $(8 \times 3.39) = 1.32$. However, since tokenization can be **executed offline and is fully parallelizable**, the overhead is negligible in practice. For instance, OverlapBPE only takes **7 minutes** to process **47.9k molecules** using 32 CPUs for virtual screening training data.

## C.2   H-MINT

**SE-(3) Equivariance.** In this section, we provide more details about the model design. At first, h-MINT follows GET's (Kong et al., 2024) architecture with 2-channel updates: One is the equivariant channel, which mainly encodes and predict the coordinates of molecules following SE-(3) symmetry. The other is the invariant channel, which mainly encodes and predicts embeddings (*i.e.*, $H$). Thus, in general, our model is an SE-(3) equivariant model. Since we only use the embedding channel in this paper, we mainly show the update and message passing for the invariant channel. However, our model can also be extended to SE-(3) scenarios like structure prediction and generation.

**Difference with GET.** We want to emphasize the difference between h-MINT and GET. GET is designed for non-overlap tokenization, while h-MINT is designed for overlap tokenization, and this induces a fundamental difference in model design. For GET, atoms and tokens are in a 1-1 mapping, making its model design simple and straightforward. For h-MINT, atoms and tokens are in a many-to-many mapping, which requires a bidirectional indexing system to convert atom-level embeddings to token-level embeddings and vice versa.

In Section 3.2, we present the embedding layer, graph construction, and Bilevel Graph Attention Layer. For the next, we complement other modules, including Bidirectional Equivariant Feed-Forward Network (FFN) and Bidirectional Equivariant Layer Normalization (LN).

**Bidirectional Equivariant FFN.** For this module, the input contains the atom-level embedding $H^l$ passed from the former module, and also the bidirectional mapping between atoms and tokens

$\mathcal{T}_{a2f}, \mathcal{T}_{f2a}$. The update is as follows:

$$H^{l,f} = \text{ScatterMean}(H^l, \mathcal{T}_{a2f}) \tag{17}$$

$$H^{l'} = \text{ScatterMean}(H^{l,f}, \mathcal{T}_{f2a}) \tag{18}$$

$$H^l \leftarrow H^l + \text{MLP}([H^l; H^{l'}; \text{RBF}(D)]), \tag{19}$$

where $H^{l'}$ can be regarded as token-enhanced atom representation, and $D$ stores the pairwise atom distances.

**Bidirectional Equivariant LN.** This module involves normalization within each input pair, since we do not use the equivariant channel. Here, we apply simple Layer Normalization to the representations.

$$H^l \leftarrow \frac{H^l - \mathcal{E}[H^l]}{\sqrt{\text{Var}[H^l] + \epsilon}} \cdot \boldsymbol{\sigma} + \boldsymbol{\mu}, \tag{20}$$

where $\boldsymbol{\sigma}$ and $\boldsymbol{\mu}$ are learnable parameters, and $\mathcal{E}[]$ and $\text{Var}[]$ are used to calculate the mean and variance of the variable.

## D    EXPERIMENTS: BINDING AFFINITY PREDICTION

### D.1    BENCHMARK DETAILS

We follow the data processing of (Somnath et al., 2021; Wang et al.) to conduct experiments on the PDBBind (v2019) Benchmark. More specifically, we use the split with a sequence identity of 30% to prevent leakage. This filtering results in 4,709 complexes, which are then split into 3,507, 466, and 490 for training, validation, and testing (Somnath et al., 2021). We directly take the baseline results from (Wang et al.; Kong et al., 2024). The main advantage of GET-PS and our model over other baselines is tokenizing small molecules into fragments. We simply remove pairs without small molecules, which results in a slightly smaller split (4436 samples) compared to the original identity30 data split (4463 samples).

### D.2    BASELINE MODELS

Here we briefly introduce the baseline methods. For the **PDBBind** benchmark, DeepDTA (Öztürk et al., 2018), Bepler and Berger's (Bepler & Berger, 2019), TAPE (Rao et al., 2019), ProtTrans (Elnaggar et al., 2022), MaSIF Gainza et al. (2020), IEConv (Hermosilla et al., 2020), Holoprot (Somnath et al., 2021), and ProNet (Wang et al.) use separate encoders for pockets and ligands. GVP (Jing et al., 2020), Atom3D (Townshend et al., 2020), and GET (Kong et al., 2024) instead use a joint encoder for pockets and ligands. Inspired by the good performance and trends in joint encoder models, we also adopt a joint encoder architecture. For the **LBA** dataset, SchNet (Schütt et al., 2018), DimeNet++ (Gasteiger et al., 2020), GemNet (Gasteiger et al., 2021) are invariant models based on invariant geometric features (*i.e.*, distance and angle). EGNN (Satorras et al., 2021), TorchMD-Net (ET) (Thölke & De Fabritiis, 2022), and LEFTNet (Du et al., 2023) preserve equivariant features and are directly implemented on 3D coordinates. MACE (Batatia et al., 2022) and Equiformer (Liao & Smidt, 2022) utilize harmonic and irreducible representations to preserve high-order equivariant features. We also include atom-level pretrained models, UniMol (Zhou et al., 2023), ProFSA (Gao et al.) and BigBind (Feng et al.). In general, all these models mainly use their invariant channel for affinity prediction, similar to GET (Kong et al., 2024) and our model; thus, we can ignore how these models deal with equivariant features in these experiments. We take the baseline results mainly from GET (Kong et al., 2024), which provides a complete comparison of all the above models in 3 representation settings: atom-level, fragment-level, and bi-level. To save space, we include each model's best representation setting only.

### D.3    IMPLEMENTATION DETAILS

We conduct experiments on 1 RTX A6000 GPU. Each model is trained with the Adam optimizer and learning rate decay. Considering the number of tokens and atoms may vary with different input

complexes, to safely and efficiently utilize the GPU memory, we implement a dynamic batch to include as many complexes as possible while not exceeding some threshold max_n_vertex. For graph construction, we use $k = 9$ for the KNN token graph, which means each token is connected with 9 nearest tokens within a complex. And we use $k = 3$ for the token-expanded atom graph, which means for atom $a_s \in f_i$, for each KNN of the token, $f_j \in \text{KNN}(f_i)$, $a_s$ will connect to 3 nearest atoms in $f_j$. We set the RBF kernel size to be 32. For baseline models, we follow the official parameter configuration. For our model, we mainly tune the learning rate ($\text{lr} \in [1e-3, 1e-4]$), final learning rate ($\text{flr} \in [1e-3, 1e-6]$), and max number of epochs ($\text{max\_epoch} \in [10, 40]$).

### D.4 ADDITIONAL EXPERIMENT ANALYSIS

**Significance of Improvements.** Since the benchmarks (PDBBind and LBA) are relatively small, to guarantee fair comparison and consistent results, we report mean and std for 3 runs in Table 1 and 2. Besides, we also conduct a significance test on the prediction results of GET, GET-PS and our model. And the results show $p$-values $< 0.005$ for these models in both PDBBind and LBA tasks. This evidence suggests that our model performs significantly better than the strong baselines, GET, and GET-PS.

**PDBBind Results Analysis.** From Table 1, we can mainly draw the following conclusions: (i) Joint encoder is generally better than separate encoders. (ii) GET outperforms other baselines by a wide margin due to its hierarchical modeling, and GET-PS performs even better. (iii) Our model consistently outperforms GET and GET-PS by a wide margin (4%-5% Pearson Correlation, and 5% Spearman Correlation). We are also surprised by the improvements, and we recognize that the improvements come from our new tokenization, OverlapBPE, and our new model h-MINT.

**LBA Results Analysis.** Table 2 compares different models with atom-level, fragment-level, and bi-level representations. We take the baseline results from (Kong et al., 2024), which compares all baselines in all 3 representation settings. In this paper, we only include the baseline results in their best representation settings to save space. From this table, we can see: (i) In general, bi-level representation is better than atom-level or fragment-level. This is reasonable because some molecular interactions happen on atoms (H-bond) and some happen on fragments ($\pi$ stacking). Thus, we believe integrating bi-level information is crucial for modeling molecular interactions. (ii) GET and GET-PS still outperform other baselines by a wide margin (2% Pearson and Spearman Correlation), which demonstrates the effectiveness of their unified representations and model design. (iii) Last, our model outperforms GET and GET-PS even more than their improvements (3% Pearson Correlation, 2% Spearman Correlation). This result again validates the effectiveness of our OverlapBPE tokenization and our new model for many-to-many mapping between atoms and tokens.

**Ablation Study of OverlapBPE and h-MINT model.** OverlapBPE tokenization and overlap-compatible hierarchical interaction model, jointly form an integrated framework to tackle the challenge of fuzzy boundaries of meaningful molecular substructures in 3D molecular interaction modeling. Meaningful comparison can only be made when they're used together, because (i) no other network architectures are available for overlapping substructures, and (ii) when non-overlapping tokenization is used, the molecular graphs for h-MINT and GET become identical. As evidence, we provide the following ablation on LBA dataset that adopts non-overlapping PS tokenizer for h-MINT in Table 7.

Table 7: Ablation Study of OverlapBPE and h-MINT on LBA.

| | RMSE ↓ | Pearson ↑ | Spearman ↑ |
|---|---|---|---|
| GET | 1.331 ± 0.008 | 0.618 ± 0.005 | 0.607 ± 0.005 |
| GET+PS | 1.312 ± 0.016 | 0.631 ± 0.011 | 0.642 ± 0.011 |
| h-MINT+PS | 1.321 ± 0.010 | 0.633 ± 0.007 | 0.641 ± 0.008 |
| GET+OverlapBPE | N/A | N/A | N/A |
| Ours (h-MINT+OverlapBPE) | **1.276 ± 0.011** | **0.660 ± 0.001** | **0.661 ± 0.001** |

As can be seen in this table, GET is not compatible with OverlapBPE. h-MINT+OverlapBPE consistently outperforms models with PS tokenizer, demonstrating a clear gain of OverlapBPE.

When non-overlap PS tokenizer is used, GET+PS and h-MINT+PS achieve similar performance as expected. The tokenizer and h-MINT architecture are both necessary to handle overlapping fragments and preserve key chemical information (atomic integrity, ionic states, chirality).

## E  EXPERIMENTS: VIRTUAL SCREENING

### E.1  DETAILS ON LOSS FUNCTION

According to LigUnity (Feng et al., 2025), we optimize a composite objective

$$\mathcal{L} \;=\; \underbrace{(\mathcal{L}_{p\to l} + \mathcal{L}_{l\to p}) + \mathcal{L}_{\text{rank}}}_{\mathcal{L}_{\text{LigUnity}}} + \lambda_{\text{mse}}\mathcal{L}_{\text{mse}}, \tag{1}$$

which extends the original loss with an additional regression term.

**Contrastive retrieval losses.** For a mini-batch of $B$ pocket embeddings $p_i$ and ligand embeddings $l_j$ we define

$$\mathcal{L}_{p\to l} = -\frac{1}{B}\sum_{i=1}^{B}\log\frac{\exp\big(\tau\,\langle p_i, l_i\rangle\big)}{\sum_{j=1}^{B}\exp\big(\tau\,\langle p_i, l_j\rangle\big)}, \qquad \mathcal{L}_{l\to p}\ \text{sym.}, \tag{2}$$

**Listwise ranking loss.** Given a pocket $i$ with $M_i$ ligands sorted by experimental affinity $\pi_1 \succ \ldots \succ \pi_{M_i}$,

$$\mathcal{L}_{\text{rank}} = -\sum_{k=1}^{M_i-1}\mu_k\,\log\frac{\exp\big(\tau\,\langle p_i, l_{\pi_k}\rangle\big)}{\sum_{t=k}^{M_i}\exp\big(\tau\,\langle p_i, l_{\pi_t}\rangle\big)}, \tag{3}$$

with positional weights $\mu_k = \frac{1}{\log(k+1)}$. However, as we used PDBBind dataset to re-train the LigUnity model, and the PDBBind dataset contains 1 to 1 pocket-ligand pairs only, the listwise ranking loss here was not functional.

**MSE loss.** Similar to the regression loss implemented in the LigUnity paper, let $\hat{a}_{ij}$ be the predicted activity for pair $(i,j)$ and $a_{ij}$ the ground truth. With $\mathcal{P}$ the positive set, $\mathcal{N}$ represents a 20 % subsample of negatives, pocket-wise weakest positive $a_{\min,i}$ and safety margin $\delta$, which was set to 2.0:

$$\mathcal{L}_{\text{mse}} = \frac{1}{|\mathcal{P}| + |\mathcal{N}|}\left(\sum_{(i,j)\in\mathcal{P}}(\hat{a}_{ij}-a_{ij})^2 + \sum_{(i,j)\in\mathcal{N}}\Big[\max\big(0,\hat{a}_{ij}-(a_{\min,i}-\delta)\big)\Big]^2\right). \tag{4}$$

Throughout this paper, we fix the weights to $\lambda_{\text{mse}} = 2$. Empirically, the extra MSE term accelerates convergence and mitigates the overfit on the training dataset.

### E.2  IMPLEMENTATION DETAILS

The implementation of virtual screening experiments includes two parts: training the baselines and finetuning our model. All the models were trained on 1 RTX 6000 GPU with 24 GB memory. For the baseline models, we retrained LigUnity (Feng et al., 2025) on the PDBBind dataset. To ensure a fair evaluation, we excluded all the samples that exist in any of the test datasets. We also followed their papers' original parameters. To evaluate, we averaged the weights of the last 3 model checkpoints. We finetuned our h-MINT model on the same PDBBind dataset. Namely, the training parameters were similar to LigUnity, with learning rate = $1e-4$, warmup ratio = 0.06, and the maximum number of ligands selected for each pocket was 16. However, we used 32-bit precision for training rather than the original 16-bit, changed the batch size to 96, and set the gradient clip to 10. We trained the model for 100 epochs initially, and observed that it converged at around the 25th epoch. Therefore, we trained it for 25 epochs and averaged the last 3 checkpoints for evaluation purposes.

### E.3  EVALUATION METRICS

We assess model performance using the following metrics:

Table 8: Zero-shot Virtual Screening on DEKOIS.

|  | AUC (%) ↑ | BEDROC (%) ↑ | EF ↑ 0.5% | 1% | 5% |
|---|---|---|---|---|---|
| LigUnity | 76.92 | 47.20 | **18.57** | 16.25 | 8.21 |
| Ours | **81.05** | **47.71** | 18.085 | **16.77** | **8.74** |

Table 9: Zero-shot Virtual Screening on JACS/Merck.

|  | $r^2$ |
|---|---|
| LigUnity | 0.173 |
| Ours | **0.216** |

- **AUC-ROC (Area Under the Receiver Operating Characteristic curve):** measures the probability that a randomly chosen active compound is ranked higher than a randomly chosen inactive one. In our paper, we use AUC to denote it.

- **BEDROC (Boltzmann-Enhanced Discrimination of Receiver Operating Characteristic):** emphasizes early recognition of actives by applying an exponential weighting to the ROC curve, controlled by a tunable parameter. Following previous works, we set the parameter to 80.5.

- **Enrichment Factor (EF):** quantifies the fold-increase in actives found among the top percentile of the ranked library relative to random selection, reported here at 0.5%, 1%, and 5%.

### E.4 Additional Results on DEKOIS, JACS/Merck, and Significance Test

We also provides comparison with LigUnity following (Feng et al., 2025) in Table 8 and 9. These results validate that h-MINT generalizes effectively across datasets and tasks, including both affinity-ranking and virtual-screening benchmarks.

To demonstrate the significance of our results, we conducted statistical significance tests for all benchmarks (DUDE, PCBA, DEKOIS, and FEP). For all comparisons between h-MINT(ours) vs. LigUnity, we obtained $p$-values < 0.005, demonstrating that the improvements are statistically significant and consistent, rather than due to random variation.

### E.5 Ablation of MSE Loss over LigUnity

To isolate the contribution of this loss, we trained LigUnity with additional MSE loss (exact same combined loss as ours) under identical settings on the PDBBind training set, as shown in Table 10. The results on DUDE and LIT-PCBA are shown in the table above, which confirms the following findings: (i). Effect of the proposed auxiliary loss. Using our proposed auxiliary loss consistently improves LigUnity across almost all metrics on both datasets. For example, on DUDE, AUC improves from 81.69 to 82.57, and BEDROC from 46.01 to 47.58. On LIT-PCBA, early-recognition metrics show noticeable gains as well. This confirms that additional loss is beneficial and strengthens the model's scoring ability. (ii). Advantage of h-MINT (LigUnity+MSE vs Ours). Our model further improves over LigUnity+MSE on most metrics. Gains are particularly clear in early-recognition measures such as BEDROC and EF0.01, which are widely regarded as key metrics for virtual screening. These results confirm that: our proposed regression loss is effective, but our h-MINT architecture delivers additional, consistent boosts beyond what the regression loss alone can offer. Thus, the comparison with LigUnity is fair, and the observed improvements come from both components of our method.

Table 10: Ablation of MSE Loss over LigUnity

| Dataset | Model | AUC (%) ↑ | BEDROC (%) ↑ | EF ↑ 0.5% | 1% | 5% |
|---------|-------|-----------|--------------|-----------|-----|-----|
| DUDE | LigUnity | 81.69 | 46.01 | 34.44 | 29.07 | 10.26 |
| | LigUnity+MSE | 82.57 | 47.58 | **35.83** | 29.77 | 10.70 |
| | Ours | **84.47** | **47.65** | 35.06 | **29.90** | **10.76** |
| LIT-PCBA | LigUnity | 57.61 | 4.34 | 4.07 | 3.04 | **2.26** |
| | LigUnity+MSE | 57.68 | 5.64 | 6.50 | 4.22 | 2.14 |
| | Ours | **57.77** | **6.21** | **7.01** | **5.20** | 2.18 |

# F    ADDITIONAL ANALYSIS AND EXPERIMENTS

## F.1    VOCABULARY ANALYSIS

We provide vocabulary statistics on LBA training set in Table 11. For all the datasets, we extract vocabularies solely from the training set, without introducing additional data.

Table 11: Vocabulary Statistics on LBA training set.

| min_freq | # tokens (basic / composite / all) | avg token size (basic / composite / all) |
|----------|-----------------------------------|------------------------------------------|
| 200 | 41 / 52 / 93 | 2.56 / 5.81 / 4.38 |
| 100 | 54 / 80/ 134 | 2.87 / 6.53 / 5.05 |
| 50 | 74 / 137 / 211 | 3.11 / 7.44 / 5.92 |
| 20 | 94 / 200 / 294 | 3.28 / 8.11 / 6.56 |
| 10 | 112 / 400 / 512 | 3.54 / 9.55 / 8.23 |

We vary the minimum frequency used by OverlapBPE, yielding vocabularies of sizes {98, 136, 212, 294}, Table 12. Performance improves from small vocabularies to around 200, where we observe the best aggregate results, and then slightly degrades at 294. Run-to-run variation is small (std $< 0.006$ across metrics), indicating stable behavior. These trends suggest a bias-variance trade-off: overly strict thresholds (very small vocab) underfit by missing informative fragments, whereas overly lax thresholds (very large vocab) admit rare or redundant fragments that increase sparsity and noise. A moderate threshold around 200 offers the best balance between coverage and denoising.

Table 12: **Vocabulary size ablation on PDBBind**. Values are obtained over three runs.

| Vocab size | Pearson | Spearman | RMSE |
|------------|---------|----------|------|
| 98 | $0.618 \pm 0.005$ | $0.601 \pm 0.006$ | $1.344 \pm 0.007$ |
| 136 | $0.615 \pm 0.002$ | $0.605 \pm 0.002$ | $1.332 \pm 0.003$ |
| 212 | $0.640 \pm 0.004$ | $0.626 \pm 0.005$ | $1.299 \pm 0.001$ |
| 294 | $0.622 \pm 0.002$ | $0.617 \pm 0.003$ | $1.327 \pm 0.002$ |

## F.2    COMPUTATIONAL OVERHEAD

OverlapBPE duplicates certain atoms during tokenization to maintain the continuity and integrity of chemical substructures. Because of these overlapping tokens, the final number of atoms becomes roughly 1.32 times the original. Despite this, both tokenization and graph construction are highly parallelizable and can be performed fully offline in preprocessing. In practice, the overhead is negligible: OverlapBPE only takes 7 minutes to process 47.9k molecules using 32 CPUs for virtual screening training data.

During training and inference, the main computational cost comes from the underlying UniMol encoder. h-MINT functions as a light-weight adapter, and the runtime difference compared with LigUnity is minimal: our training time is $\times 1.12$ than LigUnity, and our inference time is $\times 1.07$ than

LigUnity. Therefore, although h-MINT introduces a richer atom-fragment representation, the parallel and offline preprocessing ensures that the runtime during the actual virtual screening pipeline remains nearly unchanged.

## F.3 NOISE-ROBUSTNESS ANALYSIS

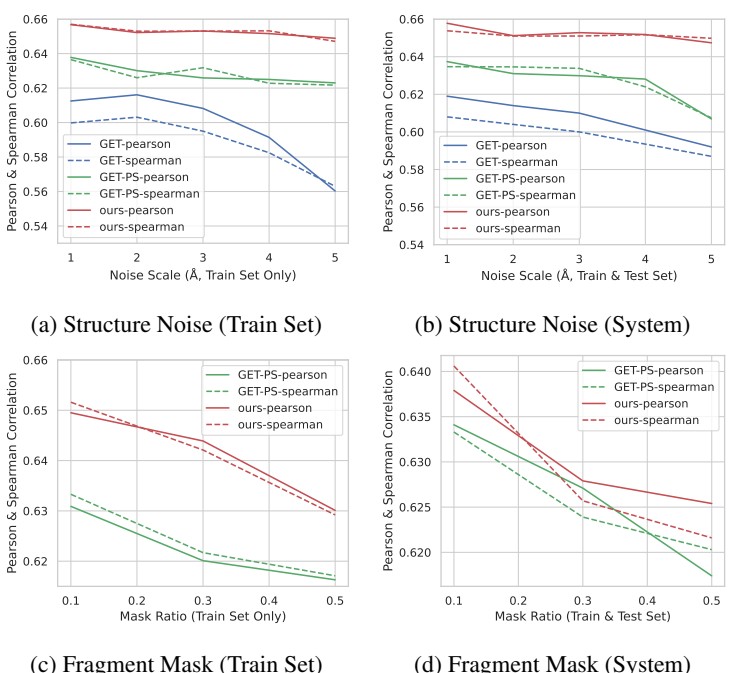

(a) Structure Noise (Train Set)  (b) Structure Noise (System)

(c) Fragment Mask (Train Set)  (d) Fragment Mask (System)

Figure 5: **Noise robustness comparison on LBA**. We report results from 3 runs.

In this section, we analyze the robustness and generalization of GET, GET-PS, and our model under different noise scales on LBA. We consider two noise settings: adding noise only to the training set for simulating scenarios when training data is of low quality or is predicted, and adding noise to both training and test sets for simulating system bias or data resolution. To investigate the effect of noise in different features, we add random noise to input structures, or randomly mask token types with some ratio[9]. The results can be found in Figure 5. In general, we can conclude that: (i) Our model is robust to structure noise even up to 5 Å. (ii) Compared with atom-level 3D structures, token types provide stronger inductive bias for binding affinity prediction, which again highlights the importance of fragment-level representations.

## F.4 FRAGMENT EMBEDDING ANALYSIS OF H-MINT

To evaluate the representation capability of the h-MINT model for molecular fragments, we first analyzed the spatial distribution of fragment embeddings using t-SNE, shown in Figure 6. The results showed that all fragments clustered into 6 distinct categories in the latent space.

Statistical analysis of fragments in each cluster, as shown in Figure 7, revealed that Cluster 5 exhibited significant chemical specificity. This cluster was predominantly enriched with functional groups containing lone electron pairs on N and O atoms, such as C=O, CNC(C)=O, and CNS(=O)(=O). These structures, acting as typical hydrogen bond acceptors, can form stable interactions with polar amino acids (e.g., the hydroxyl group of serine or the guanidinium group of arginine) in protein binding pockets. This ability to capture key chemical features enables h-MINT to more sensitively identify structural determinants affecting binding affinity, leading to superior performance over existing sequence- or graph-based baseline models on the binding affinity prediction and virtual screening tasks.

---

[9]Since GET only uses atom representations for molecules, we only compare GET-PS with ours in (c)-(d).

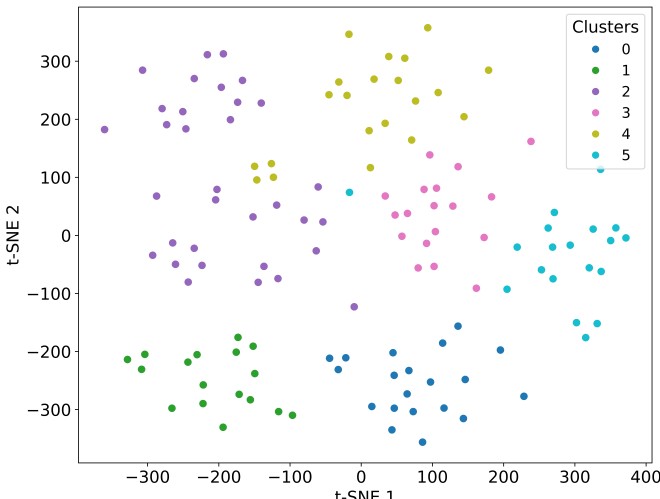

Figure 6: **t-SNE Visualization of Fragment Embeddings.**

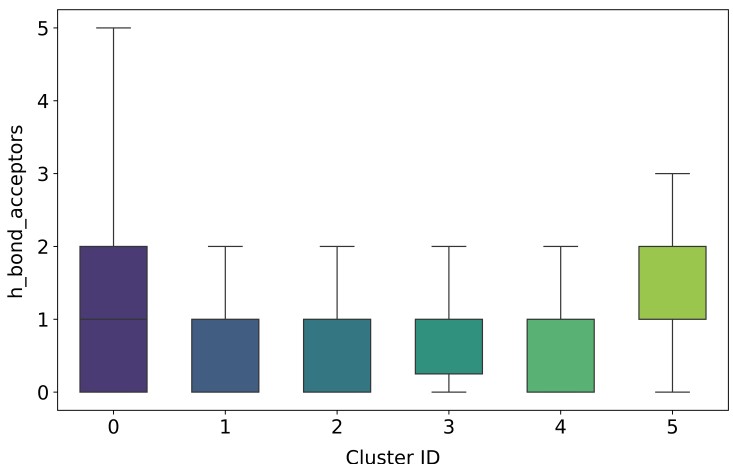

Figure 7: **H-bond acceptors distribution across clusters.**

### F.5 ADDITIONAL EXPERIMENTS ON MOLECULAR PROPERTY PREDICTION

We include 3 property prediction tasks from MoleculeNet. The baseline follows MoleculeNet directly, which extracts ECFP features and trains XGBoost with grid search. The ECFP features are chosen from 128-bit, 512-bit, 1024-bit and 2048-bit according to dataset. We augment the ECFP features with bag-of-word features extracted from OverlapBPE and train the same XGBoost. The results are in Table 13. The significant improvements in prediction error confirm that OverlapBPE provides discriminative representations for molecules.

Table 13: Molecular Property Prediction Benchmarks from MoleculeNet.

| RMSE | ESOL ↓ | FreeSolv ↓ | Lipo ↓ |
|------|--------|------------|--------|
| ECFP | 1.5668 | 3.9498 | 0.8875 |
| ECFP + OverlapBPE | **1.2972** | **3.3409** | **0.8270** |

### F.6 EXAMPLE TOKENS AND CHEMICAL INSIGHTS

By construction, OverlapBPE induces a hierarchical organization of fragments: basic tokens correspond to chemically primitive units (atoms, bonds, individual rings), while composite tokens capture larger patterns such as fused ring systems and side chains. In this section, we analyze how the learned fragments align with standard functional chemistry.

We provide a list of the top 100 fragments mined from the LBA training set in Fig. 8, and we are able to obtain many chemically and biologically meaningful subunits or motifs. We emphasize that this motif-mining procedure is fully automatic, based solely on the data distribution, and leverages no prior chemical or biological knowledge. We analyze and categorize the mined fragments into the following chemically or biologically meaningful parts.

**Chemical functional groups.** A functional group is a specific group of atoms or bonds within a molecule that is responsible for its characteristic chemical properties and reactions. Representative functional groups mined include: *carboxyl, phosphate, amide (peptide bond), benzyl, secondary amino, tertiary amino, and quaternary amino (ammonium).* These functional groups are small chemical subunits that were often selected by hand in previous functional-group-based tokenization but can be easily recovered with our approach.

**Biomolecule subunits.** Compared to simple chemical functional groups, biomolecules such as proteins, DNA, RNA, and polysaccharides are significantly larger. Yet, the monomers that constitute these biomolecules exhibit characteristic patterns, such as amino acids, nucleotides, and saccharides. Indeed, we observe a considerable number of biologically meaningful fragments in our vocabulary: *peptide bond, adenine, pyranose, deoxyribose*, and, most notably, the whole nucleotide *adenosine* monophosphate, which consists of an adenine, a ribose, and a phosphate. To the best of our knowledge, none of the existing fragmentation approaches has been able to mine such large subunits while preserving their biological significance.

**Drug subunits.** Remarkably, we also observed that our approach could mine large chemical subunits that are common motifs in small-molecule drugs. These include: *adamantane*, a 10-carbon tricyclic motif with a highly symmetric fused four hexane rings, occurring in some antiviral drugs; *sulfonamide*, in the antibacterial drug sulfanilamide; and most notably, the 28-heavy atom *sulfonamide protease inhibitor motif* in HIV drugs. Indeed, our approach automatically mined this large subunit, demonstrating its effectiveness at capturing the biochemical roles of many motifs.

## G THE USE OF LARGE LANGUAGE MODELS

We employ large language models exclusively for language editing, which is limited to polishing text to improve readability. No language models contributed to the development of research ideas, analysis, models, or interpretation of results.

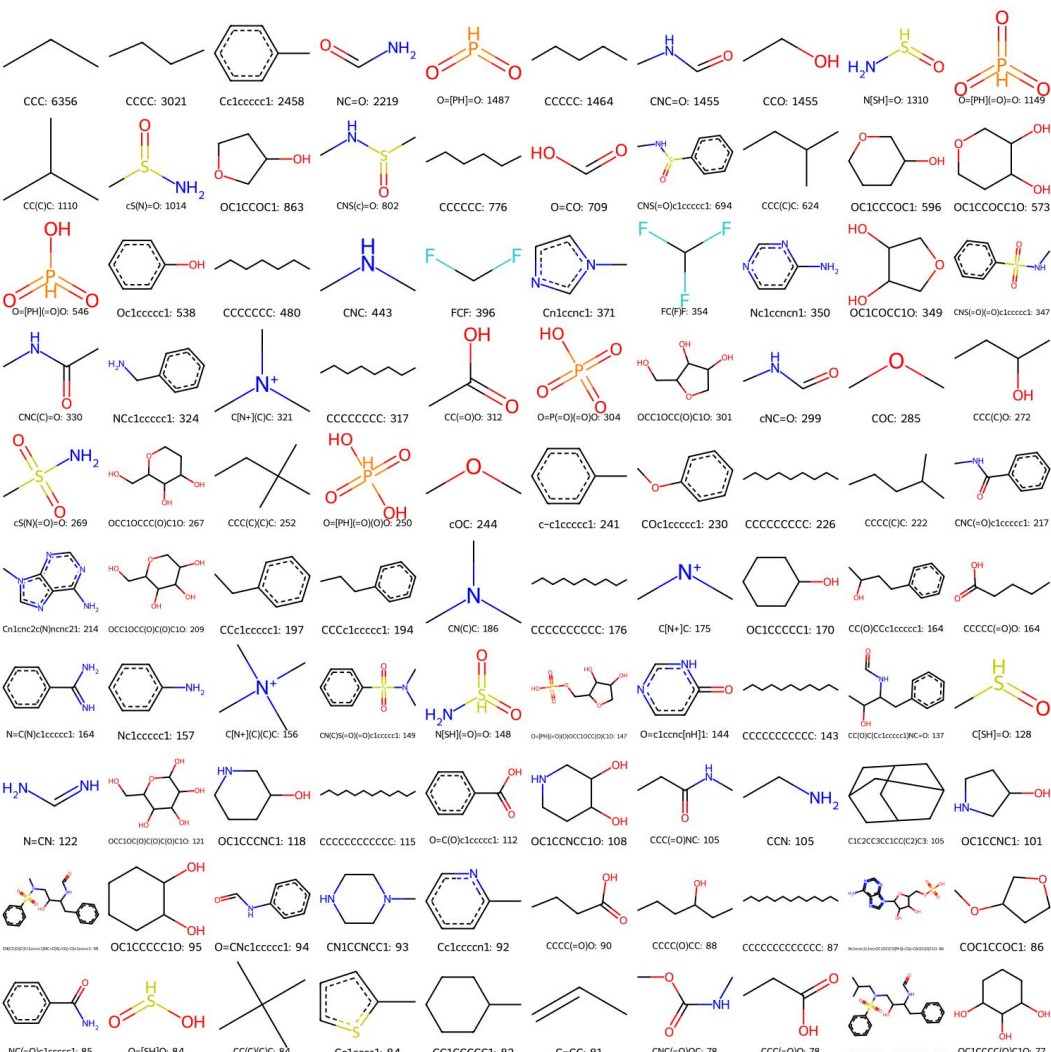

Figure 8: Top-100 composite tokens from LBA vocabulary.

