# OpenReview forum: "h-MINT: Modeling Pocket-Ligand Binding with Hierarchical Molecular Interaction Network"
_ICLR.cc/2026/Conference — ICLR 2026 Poster_

### Official Review · Reviewer_e8QX · 2025-10-21

**Soundness:** 3
**Presentation:** 3
**Contribution:** 2
**Rating:** 8
**Confidence:** 3

**Summary:**

The authors propose a method for biomolecular interaction modelling (protein-ligand binding, specifically). This method has two novel components: 1) OverlapBPE tokenization, which is a novel tokenization technique building upon BPE, where different tokens may overlap each other, capturing more complex molecular substructures, like rings and their functionalizations; 2) h-MINT, a hierarchical SE(3)-equivariant graph neural network built to exploit both the 3D information about protein-ligand pockets and the fragment information derived from OverlapBPE.

The results show that the method improves with regards to relevant baselines

**Strengths:**

1. The OverlapBPE tokenization is an elegant way of capturing molecular substructures, while preserving functional units.
2. The h-MINT, although not particularly novel on its own (as it is a hierarchical GNN), is a reasonable choice for exploiting the information from the protein-ligand binding poses and the fragment information derived from OverlapBPE.
3. The results show the advantage of the method.
4. The evaluation describes the deviation between multiple experiments which provides a necessary idea of the significance of the differences with regards to other methods.

**Weaknesses:**

1. The choice of evaluation benchmarks might lead to undiagnosed overfitting, as the controls for data leakage focus on the protein targets, rather than the molecular ligands. This has been shown to be problematic in some cases. As this problem occurs both on the method and the baselines is not a critical problem, but it is a limitation that should be acknowledged.

**Questions:**

None

---

> ### Author Response · Authors · 2025-11-25
>
> We sincerely thank the reviewer for the positive and encouraging assessment of our work. We are pleased that the reviewer considers OverlapBPE an **elegant** substructure-preserving tokenizer and finds h-MINT to be a **well-justified** hierarchical architecture. We also appreciate the reviewer’s positive remarks on our **experimental improvements** and our careful reporting of deviation from multiple runs.
>
> **(W1)** Discussion on bias in protein-based data splits
>
> Thank you for highlighting this important perspective on dataset splitting. We fully agree that ligand-level biases can influence data-driven models, and we appreciate the opportunity to clarify our position.
>
> - *As a **data-driven** method, our approach inevitably inherits ligand-level biases present in the datasets*, e.g., ligands that naturally appear more frequently or have privileged scaffolds in affinity prediction tasks. This is a known issue in many public benchmarks and is not specific to our method.
> - Following reviewer gELf’s suggestion, we have supplemented our evaluation with additional molecular property prediction tasks from MoleculeNet, all of which use **scaffold splits**, thereby reducing ligand-level leakage and demonstrating that our tokenizer and model also generalize well under stricter splitting strategies.
> - Importantly, we acknowledge that **the commonly used affinity benchmarks control data leakage at the protein level but do not fully eliminate ligand-level overlap**.
> - To address the reviewer’s concern more concretely, we have added a discussion of stricter ligand-level controls such as **scaffold-based ligand splitting, Murcko framework filtering, and near-duplicate removal (e.g., Tanimoto > 0.8)** as promising directions for future work. We plan to evaluate our method under these more stringent settings in subsequent versions.
> - Finally, we note that ligand-level duplication can introduce bias, but it does not dominate the evaluation of our method, as binding affinity prediction is strongly influenced by protein–ligand interaction patterns rather than ligand identity alone. Nonetheless, we fully agree that this is a limitation worth acknowledging and add further discussion in revised version.

---

> > ### Comment · Reviewer_e8QX · 2025-11-25
> >
> > I acknowledge the authors rebuttal and appreciate the changes they have proposed. As my recommendation was already to accept the paper, I will not modify it.

---

> > > ### Author Response · Authors · 2025-11-25
> > >
> > > Dear Reviewer e8QX,
> > >
> > > Thank you for your careful evaluation and for maintaining your recommendation. We sincerely appreciate your constructive feedback and support throughout the review process!

---

### Official Review · Reviewer_gELf · 2025-10-30

**Soundness:** 3
**Presentation:** 2
**Contribution:** 2
**Rating:** 2
**Confidence:** 4

**Summary:**

This paper introduces a novel framework for modeling protein-ligand interactions, for the purpose of improving performance on drug discovery tasks like binding affinity prediction and virtual screening. The authors identify that existing molecular representation methods, particularly those based on fragmenting molecules into disjoint sets, often fail to preserve crucial chemical information such as aromaticity, chirality, and ionic states.  A new molecular tokenization algorithm is proposed. Unlike prior methods, OverlapBPE allows fragments to overlap, better preserving the local chemical context and molecular substructures. The authors also developed a hierarchical Molecular Interaction Network for interaction prediction. Extensive experiments are conducted on several benchmarks.

**Strengths:**

1. The idea of OverlapBPE is novel and well-motivated. It effectively addresses a clear limitation in prior work.

2. The experiment is extensive. The authors evaluate their method on four different datasets.

3. The paper is very well-written and easy to follow.

**Weaknesses:**

1. The OverlapBPE is primarily for molecular representation. How does it perform on molecular tasks? It seems to me that it is more naturel to test OverlapBPE on molecular tasks than the ligand-protein interaction task.

2. The novelty of h-MINT is limited. The core architecture of the h-MINT model is heavily built upon existing equivariant graph transformers. The paper states that h-MINT follows GET's architecture. What are the precise architectural changes made for handling the overlapping tokens? Are the underlying equivariant transformer blocks identical to those in GET?

3. There is no study of the characteristics of the fragment vocabulary, such as the size and statistical information, the hierarchical structure, and the correspondence to common functional groups; thus, although the motivation is good, little chemical insight is presented.

4. What is the training corpus to compute the token frequency? Does it affect the vocabulary?

5. The contributions of the tokenization and the h-mint model are convoluted. There is no ablation to show the effect of the tokenization clearly.

6. In the experiments of VS, DrugCLIP and LigUnity are re-trained with PDBBind. Why do they have such a big difference in performance? The only difference is the ranking loss. The ranking loss in LigUnity needs samples of the same protein and different ligands, which is not common in PDBBind. How many such samples are there in PDBBind?

7. No code available. The reproducibility statement is weak.

**Questions:**

See Weaknesses

---

> ### Author Response · Authors · 2025-11-25
>
> We thank the reviewer for acknowledging our innovation and motivation in methodology, as well as representation and experimental results.
>
> **(W1) (New benchmark)** OverlapBPE's performance on molecular tasks
>
> We agree that OverlapBPE could also benefit monomer molecular tasks. However, our main motivation comes from the observation that fragment-level interaction patterns (e.g., π–π stacking, cation–π interactions, and electron-delocalization in hydrogen bond), are crucial yet largely overlooked in prior work on molecular interaction modeling. Capturing these localized chemical environments, which depend on substructure context rather than isolated atoms, motivates the use of overlapping fragment tokens for molecular interaction.
>
> - **Our HTS experiments**
>
>     We'd also like to point out that **we indeed included a molecular task with our HTS experiments**. Table 4 in our main text reports results on PubChem **HTS assays**, where only the small-molecule information is used for activity prediction. This is a challenging property prediction task, and our method achieves **clear improvements against multiple advanced GNN baselines**, particularly with OverlapBPE's chirality-aware tokenization strategy. Moreover, OverlapBPE naturally extends to complex molecular regimes such as **macrocycles, non-canonical amino acids, cyclic and D-peptides**, which can hardly be represented by non-overlap tokens or residue-level tokens.
>
> - **Additional Property-prediction benchmarks**
>
>     To further demonstrate OverlapBPE's contribution to molecular tasks, following your suggestion, we include 3 property prediction tasks from MoleculeNet. The baseline follows MoleculeNet directly, which extracts ECFP features and trains XGBoost with grid search. The ECFP features are chosen from 128-bit, 512-bit, 1024-bit and 2048-bit according to dataset. We augment the ECFP features with bag-of-word features extracted from OverlapBPE and train the same XGBoost. The results are as follows:
>
>     | RMSE | ESOL ↓ | FreeSolv ↓ | Lipo ↓ |
>     | --- | --- | --- | --- |
>     | ECFP | 1.5668 | 3.9498 | 0.8875 |
>     | ECFP + OverlapBPE | **1.2972** | **3.3409** | **0.8270** |
>
>     The significant improvements in prediction error confirm that OverlapBPE provides discriminative representations for molecules.
>
>
> These results together confirm that OverlapBPE is not only effective for ligand-protein interaction tasks but also provides **meaningful and consistent benefits** across **pure molecular tasks**, validating its utility as a general molecular representation method.
>
> **(W2/W5)** Novelty of h-MINT and ablation for isolating tokenization and modeling.
>
> Thank you for giving us the opportunity to elaborate on the architectural innovations in h-MINT. We first emphasize that h-MINT is not a drop-in reuse of GET: the GET architecture hard-codes the disjoint-set assumption and fails to support the many-to-many atom-fragment mapping required in our proposed OverlapBPE tokenization. h-MINT is purposedly designed to support our overlap tokenization, and involves non-trivial changes in both the **graph construction** and the **message-passing blocks**. We detail these differences in the table below:
>
> |  | GET | h-MINT |
> | --- | --- | --- |
> | **Graph representation** |  |  |
> | Atom-fragment mapping | Models the complex as a graph of disjoint atom sets with a strict 1-many mapping between fragments and atoms. Cannot directly accommodate overlaps | Explicitly allows fragments to share atoms (e.g., overlapping aromatic rings), which induces a many-to-many mapping between atoms and fragment tokens |
> | Handle disconnected graphs (e.g., ionic states for salt bridge) | No | Yes |
> | Handle overlap fragments | No | Yes |
> | **Graph construction** |  |  |
> | Fragment distance | The minimum distance between atom pairs from 2 fragments | Shared atoms are excluded when computing inter-fragment distances, which prevents excessive fragment-level connectivity and eliminates spurious signals like “2 fragments are superimposed” |
> | Atom–fragment adjacency matrix | No. Each atom only corresponds to a single fragment | Yes. Maintain the reverse index of atom → fragments for message update |
> | Atom-level interactions | Dense interaction | An atom may contribute differently across overlapping fragments. Our formulation preserves fragment-specificity and supports coherent global aggregation. |

---

> > ### Author Response · Authors · 2025-11-25
> >
> > (cont'd)
> >
> > |  | GET | h-MINT |
> > | --- | --- | --- |
> > | **Model architecture** |  |  |
> > | Bilevel Attention | Fragment → atom update is injective. every atom receives a unique message from a single fragment | Use the global atom index to aggregate multiple fragment → atom messages into a single atom update |
> > | Feed-forward | Single representation and relative position for each atom | Syncronize global atom representation as well as preserve local relative position in fragments |
> > | LayerNorm | Single representation and relative position to compute mean and variance for each atom | Multiple local representations are aggregated into a single global representation before computing mean and variance to preserve equivariance and numerical stability in many-to-many setting.
> >
> > We appreciate your suggestion on ablation to isolate contributions from the tokenizer and model design. However, we reiterate that our two technical contributions, OverlapBPE tokenization and overlap-compatible hierarchical interaction model, **jointly form an integrated framework** to tackle the challenge of fuzzy boundaries of meaningful molecular substructures in 3D molecular interaction modeling. Meaningful comparison can only be made when they're used together, because (1) no other architectures are available for overlapping substructures, and (2) when non-overlapping tokenization is used, the molecular graphs for h-MINT and GET become identical. As evidence, we provide the following ablation on LBA dataset that adopts non-overlap PS tokenizer for h-MINT:
> >
> > | Model | RMSE | Pearson | Spearman |
> > | --- | --- | --- | --- |
> > | GET | 1.331 ± 0.008 | 0.618 ± 0.005 | 0.607 ± 0.005 |
> > | GET+PS | 1.312 ± 0.016 | 0.631 ± 0.011 | 0.642 ± 0.011 |
> > | h-MINT+PS | 1.321 ± 0.010 | 0.633 ± 0.007 | 0.641 ± 0.008 |
> > | GET+OverlapBPE | N/A | N/A | N/A |
> > | Ours (h-MINT+OverlapBPE) | **1.276 ± 0.011** | **0.660 ± 0.001** | **0.661 ± 0.001** |
> >
> > As shown above, GET is not compatible with OverlapBPE. OverlapBPE+hMINT consistently outperforms models with PS tokenizer, demonstrating a clear gain of OverlapBPE. When non-overlap PS tokenizer is used, GET-PS and h-MINT-PS achieve similar performance as expected. The tokenizer and h-MINT architecture are both necessary to handle overlapping fragments and preserve key chemical information (e.g., substructure integrity, ionic states, chirality), and their contribution shall not be viewed in isolation.
> >
> > **(W3):** Characteristics of OverlapBPE vocabulary and chemical insights.
> >
> > Thank you for the great suggestion. Our goal with OverlapBPE is precisely to construct a fragment vocabulary that encodes chemically meaningful structure rather than arbitrary graph partitions. While the main text focuses on the modeling and performance aspects due to limited space, **we have included several representative cases and statistics of the mined fragments** in the appendix D.
> >
> > We now provide additional analysis of the vocabulary’s properties and chemical semantics as follows.
> >
> > - **Statistical properties and vocabulary size**
> >
> >     In the revised version, we now make the statistical characterization of the fragment vocabulary more explicit (built upon the LBA training set, 3.5k complexes). First, Section 3.1 describes how OverlapBPE builds the vocabulary from “basic tokens” (single atoms, bonds, and rings) and then iteratively discovers higher-order composite fragments via frequency-based merges. This defines a natural hierarchy over fragments, from simple moieties to larger scaffolds. In Appendix D.4 and Table 7, we further report an ablation over vocabulary sizes obtained by varying the minimum frequency threshold. This analysis shows a clear bias-variance trade-off: very small vocabularies underfit by missing informative fragments, while very large vocabularies introduce rare, noisy fragments and slightly degrade performance, with a moderate vocabulary (≈200 tokens) achieving the best overall results.
> >
> >     | min_freq | # tokens (basic / composite / all) | avg token size (basic / composite / all) |
> >     | --- | --- | --- |
> >     | 200 | 41 / 52 / 93 | 2.56 / 5.81 / 4.38 |
> >     | 100 | 54 / 80/ 134 | 2.87 / 6.53 / 5.05 |
> >     | 50 | 74 / 137 / 211 | 3.11 / 7.44 / 5.92 |
> >     | 20 | 94 / 200 / 294 | 3.28 / 8.11 / 6.56 |
> >     | 10 | 112 / 400 / 512 | 3.54 / 9.55 / 8.23 |

---

> > > ### Author Response · Authors · 2025-11-25
> > >
> > > (cont'd)
> > >
> > > - **Hierarchical structure and correspondence to functional groups.**
> > >
> > >     By construction, OverlapBPE induces a hierarchical organization of fragments: basic tokens correspond to chemically primitive units (atoms, bonds, individual rings), while composite tokens capture larger patterns such as fused ring systems and side chains. We have added clarifications and examples in the appendix that illustrate how a ligand is decomposed and merged across these levels. Beyond this structural hierarchy, we also analyze how the learned fragments align with standard functional chemistry. In Appendix D.3, we cluster fragment embeddings learned by h-MINT and find that multiple clusters are enriched in canonical functional groups. For example, carbonyls, amides, and sulfonamides that act as typical hydrogen-bond acceptors.
> > >
> > >     We provide a list of the top 100 fragments mined from the dataset in Appendix D, and we are able to obtain many chemically and biologically meaningful subunits or motifs. We emphasize that this motif-mining procedure is **fully automatic,** based solely on the data distribution, and leverages **no prior chemical or biological knowledge**.
> > >
> > >     We analyse and categorize the mined fragments into the following chemically or biologically meaningful parts.
> > >
> > >     - **Chemical functional groups.** A functional group is a specific group of atoms or bonds within a molecule that is responsible for its characteristic chemical properties and reactions. Representative functional groups mined include: *carboxyl, phosphate, amide (peptide bond), benzyl, secondary amino, tertiary amino, and quaternary amino (ammonium)*. These functional groups are small chemical subunits that were often selected by hand in previous functional-group-based tokenization but can be **easily recovered with our approach**.
> > >     - **Biomolecule subunits**. Compared to simple chemical functional groups, biomolecules such as proteins, DNA, RNA, and polysaccharides are significantly larger. Yet, the monomers that constitute these biomolecules exhibit characteristic patterns, such as amino acids, nucleotides, and saccharides. Indeed, we observe a considerable number of biologically meaningful fragments in our codebook: *peptide bond, adenine, pyranose, deoxyribose,* and, most notably, the whole nucleotide *adenosine monophosphate, w*hich consists of an adenine, a ribose, and a phosphate. To the best of our knowledge, **none of the existing fragmentation approaches has been able to mine such large subunits while preserving their biological significance**.
> > >     - **Drug subunits**. Remarkably, we also observed that our approach could mine large chemical subunits that are common motifs in small-molecule drugs. These include: *adamantane,* a 10-carbon tricyclic motif with a highly symmetric fused four hexane rings, occurring in some antiviral drugs; *sulfonamide*, in the antibacterial drug sulfanilamide; and most notably, the 28-heavy atom *sulfonamide protease inhibitor motif* in HIV drugs. Indeed, our approach automatically mined this large subunit, demonstrating its **effectiveness at capturing the biochemical roles of many motifs**.
> > > - **Chemical insights: ring integrity and charged fragments.**
> > >
> > >     As shown in the case study in **Figure 3**, our fragment vocabulary exhibits clear chemical properties. Specifically, the tokenization strategy (1) **preserves ring integrity**, ensuring that multi-ring and aromatic structures are not broken across fragments, thereby maintaining essential chemical structural information; and (2) **retains charge states** such as [N+], which allows the model to correctly capture interactions like the **π-cation interactions** demonstrated in the figure. These observations indicate that the vocabulary carries meaningful chemical insights beyond its statistical characteristics, and we have made this clearer in the revised manuscript.
> > >
> > > **(W4)** Training corpus for token frequency and its effect
> > >
> > > - For PDBBind and LBA, we build a vocabulary from the LBA training data. For VS, the vocabulary comes from the PDBBind training set. And for HTS, we build vocabularies based on each assay.
> > > - For a fair comparison, we construct the vocabulary exclusively from the training set without incorporating any external molecular libraries. Nevertheless, as a data-driven approach, our method can naturally benefit from large public datasets such as PubChem, ChEMBL, and ZINC. We leave the exploration of such extensions to future work.
> > > - Overall, the quality and size of the vocabulary are primarily determined by the quality and scale of the data. As a general and flexible approach, our method can be readily applied to a wide range of datasets and applications.

---

> ### Author Response · Authors · 2025-11-25
>
> **(W6)** In the experiments of VS, DrugCLIP and LigUnity are re-trained with PDBBind. Why do they have such a big difference in performance? The only difference is the ranking loss. The ranking loss in LigUnity needs samples of the same protein and different ligands, which is not common in PDBBind. How many such samples are there in PDBBind?
>
> Thank you for catching this unexpected behavior. We’d like to elaborate on our findings regarding the reproduction of DrugCLIP:
>
> - In our virtual screening experiments with PDBBind, all models are trained in a 1-pocket-to-1-ligand setting, different from the hit-to-lead task where multiple ligands are provided. Therefore, the ranking loss in LigUnity has close to zero effect due to the lack of available ranking tuples.
> - We trained DrugCLIP and LigUnity using the official GitHub code and strictly following the hyperparameters reported in the original paper, and selected the best checkpoint using validation set, which is standard practice in machine learning and aligns with the official implementation. After further investigation, we discovered a potential overfitting issue with DrugCLIP's official recommended training parameter (200 epochs).
> - To analyze this, we evaluated checkpoints across multiple epoch budgets (20–200).  We observed that DrugCLIP achieves its peak performance for DUDE around 50 epochs, whereas PCBA is less sensitive. For fairness, we also inspected the checkpoints for LigUnity and h-MINT. Neither showed observable overfitting, so our reported results remain unaffected. With this adjustment, DrugCLIP performs similarly to LigUnity on DUDE, **consistent with the findings in the LigUnity Paper**.
>
>
>     | Dataset | Model | AUC | BEDROC | EF0.005 | EF0.01 | EF0.05 |
>     | --- | --- | --- | --- | --- | --- | --- |
>     | DUDE | DrugClip (paper param) | 78.08 | 37.22 | 27.57 | 20.83 | 8.41 |
>     |  | DrugClip (best ckpt) | 81.39 | 45.96 | 34.27 | 29.01 | 10.18 |
>     |  | LigUnity | 81.69 | 46.01 | 34.44 | 29.07 | 10.26 |
>     |  | Ours | **84.45** | **47.64** | **35.06** | **29.91** | **10.76** |
>     | PCBA | DrugClip | **58.15** | 4.12 | 4.11 | 3.08 | **2.27** |
>     |  | LigUnity | 57.61 | 4.34 | 4.06 | 3.03 | 2.25 |
>     |  | Ours | 57.77 | **6.27** | **7.01** | **5.20** | 2.18 |
> - We would like to further emphasize that, even after correcting for DrugCLIP's overfitting and updating Table 3 and Table 6 accordingly (and also shown in the table above), h-MINT still consistently outperforms both LigUnity and DrugCLIP across virtual screening and affinity-ranking benchmarks. Thus, the advantages of our method remain robust
>
> **(W7)** No code available.
>
> We would like to assure that reproducibility has been our top priority, and we have made extensive efforts on that, including disclosure of full details of model architecture, tokenization algorithm, training parameters, data preprocessing procudures and evaluation protocols. All datasets used in our experiments (PDBBind, LBA, DUD-E, etc.) are standard public benchmarks with clearly specified splits and filtering criteria. We also used the official repo for the reproduction of baselines (LigUnity, DrugCLIP, GET, etc.) following their recommended parameters. We have included a reproducibility statement as recommended by the guidelines. Concretely, Section 3.1 describes the OverlapBPE tokenization algorithm, and Section 3.2 details the h-MINT architecture, including the embedding layers, hierarchical graph construction, and bilevel attention mechanism. Sections 4.1 and 4.2 define the binding-affinity and virtual screening tasks, datasets, and evaluation metrics. Appendix B.1 and B.2 provide PDBBind/LBA preprocessing and data splits as well as full training and hyperparameter settings,  Appendix C.3 gives implementation details for the virtual screening experiments.
>
> While we adhered to the ICLR guidelines, which do not mandate code submission for review, **we are fully committed to open-sourcing our code and pre-trained models upon acceptance.** In the interim, if there are specific implementation details the reviewer wishes to verify, we are happy to provide an anonymized repository.
>
> We would be happy to further clarify any remaining questions or discuss any points that may still be unclear. If our responses have properly addressed your concerns, we kindly hope you will consider reflecting this in your final assessment.

---

### Official Review · Reviewer_EmZ6 · 2025-10-30

**Soundness:** 3
**Presentation:** 3
**Contribution:** 3
**Rating:** 4
**Confidence:** 5

**Summary:**

The paper proposes an approach for tokenization of molecular graphs by considering overlapping tokens. It starts with frequent basic subgraphs treated as words in  an initial vocabulary. Adjacent subgraphs are merged to  form new tokens,  ovelapping words are allowed in the context. New tokens' frequency is calculated across the database and most frequent new tokens are kept.
The tokenized graphs are transformed into new graphs whereas the nodes are subgraphs and edge are overlapping information. The authors use an equivariant  GNN to transform the given subgraphs for predicting binding affinity with target. The experimental results carried out on two standard benchmarks and comparison was done to compare the proposed approaches with baselines regarding equivariant  GNN used for neural potential learning and other variation of all atoms GNN. The proposed approach outperformed the given approaches with a significant margin.

**Strengths:**

The idea is interesting, the results look significant compared to related baselines.

**Weaknesses:**

I have a concern on the chosen baselines. It is well known that  representation learning approaches is behind  molecular fingerprint on molecular property prediction tasks. Molecular fingerprint such as Morgan Fingerprints, ERP, Avalon available in the RDKit tools can summary important substructure and functional groups in a small molecule and being better or comparative to other molecular representation learning approaches (see the leaderboard for ADMET drug property prediction by TDC).

The idea provided in this paper is not new if considering molecular fingerprint where overlappping subgraphs or functional graphs are considered as a fingerprint of the molecules. The new part may concern the preserve of the connectivity between substructure in the graphs but it is unclear whether that information is useful. I would suggest the authors to perform additional experiments whereas fingerprints are used as feature representation of ligands and ESMas feature representation of proteins, then use simple approach such as lightgbm on the top of the provided features to predict binding affinity. That would be a strong confirmation of the significance of the proposed methods compared to existing fingerprint based approaches.

**Questions:**

Could you please compare your approach to simple baselines: taking ESM-2 as representation of protein, MorganFingerprint , ERP and Avalon as combined representation of ligands, run HPO in lightgbm models to find best set of hyperparameters of lightGBM and report the results of that baseline approaches in your paper?
If the experiments show significance results w.r.t to that baseline I am happy to raise my score.

---

> ### Author Response · Authors · 2025-11-24
>
> Thank you for recognizing the **novelty** of our method and the **strength of our experimental results**. We hope that the additional experiments we have provided adequately address your questions.
>
> **W1 & Q1: (New baseline)** Concerns about **innovations** and **improvements** over suggested baseline (ESM-2+figerprint).
>
> We thank the reviewer for this insightful comment, and we'd like to elaborate on our methodological novelty and its key distinctions from molecular fingerprints.
>
> Innovations:
>
> - Our approach is **data-driven** and automatically learns task-relevant molecular substructures and their spatial relationships from training data, going beyond fixed, handcrafted fingerprints. Although fingerprint-based methods can preserve overlapping molecular substructures, they primarily rely on **fixed, hand-crafted rules or functional groups**. The novelty of our OverlapBPE lies in its data-driven formula for mining statistically frequent and chemically important patterns, which is the first of its kind among all data-driven fragmentation methods (e.g., PS tokenizer used in GET).
> - Our method builds a **bi-level pocket–ligand interaction graph** that preserves
>     - both atom- and substructure-level connectivity, which enables bi-level interaction and message passing.
>     - The **3D geometry** of the **protein-ligand interaction interface**, not just the 2D intra-ligand topology. Such information is crucial for **interaction-centric tasks** such as binding affinity prediction and structure-based virtual screening. Classical fingerprints summarize only the *intrinsic* substructures of an isolated ligand into a fixed 1D vector, discarding **the binding pocket, residue environment, or spatial contacts**. As a result, they cannot distinguish ligands that are topologically similar but bind in **different poses**, contact **different residues**, or form **different interaction patterns**, all of which can drastically affect binding energy.
>     - Since our main focus is interaction modeling in this paper, we argue that such connectivity and spatial contacts are crutial for these tasks.
>
> Following your kind suggestion, we have added an experiment where we use **ESM-2 embeddings** as protein features and a concatenation of **MorganFingerprint, ERP, and Avalon** as ligand features, and train a **LightGBM** model on top. We perform grid-search hyperparameter optimization and report the mean ± std over three random seeds on PDBBind. The results are:
>
> | Model | RMSE | Pearson | Spearman |
> | --- | --- | --- | --- |
> | ESM-2 + Fingerprint | 1.537 ± 0.001 | 0.455 ± 0.013 | 0.433 ± 0.009 |
> | Atom3D-GNN | 1.601 ± 0.048 | 0.545 ± 0.027 | 0.533 ± 0.033 |
> | GET | 1.430 ± 0.007 | 0.586 ± 0.001 | 0.575 ± 0.002 |
> | GET-PS | 1.387 ± 0.015 | 0.601 ± 0.002 | 0.582 ± 0.005 |
> | **Ours** | **1.295 ± 0.001** | **0.640 ± 0.002** | **0.625 ± 0.002** |
>
> It can be seen that although the fingerprint baseline performed better than Atom3D-GNN, it underperformed GET and our methods significantly,  despite using strong protein features (ESM-2) and a variety of ligand fingerprints with LightGBM+HPO. We believe the main reason is that this baseline lacks explicit 3D information and pocket-ligand interaction geometry, especially the binding pose, which are explicitly modeled in our fragment graph and bi-level interaction architecture.
>
> Finally, we view our method as **complementary** to traditional fingerprints rather than contradictory: fingerprints encode domain knowledge in a fixed feature space, while OverlapBPE + h-MINT provides a flexible, data-driven interaction-centric representation that can adapt as more structural data become available. More importantly, these two approaches can be synergistically combined (e.g., by adding fingerprint as additional node features to h-MINT). We will clarify this distinction and include the new ESM-2 + fingerprint + LightGBM baseline in the revised manuscript.
> Thank you for your great suggestion again, and we sincerely hope that our additional results have confirmed the significance of our method as you suggested.

---

### Official Review · Reviewer_vwks · 2025-10-30

**Soundness:** 2
**Presentation:** 3
**Contribution:** 2
**Rating:** 4
**Confidence:** 4

**Summary:**

This paper proposes OverlapBPE, an overlapping, chemically consistent fragment tokenization that preserves aromatic systems, chirality, and ionic states, and h-MINT, a hierarchical, SE(3)-equivariant interaction network that couples atom–fragment representations via bilevel attention and token-expanded geometric edges. On PDBBind and LBA, h-MINT improves binding-affinity prediction over baselines; on DUD-E/LIT-PCBA it yields better zero-shot virtual screening metrics.

**Strengths:**

1. The proposed OverlapBPE tokenization effectively preserves crucial chemical information.

2. The h-MINT architecture enables many-to-many atom–fragment interactions via bilevel attention and equivariant message passing.

**Weaknesses:**

1. The tokenizer and architecture are co-developed, but their effects are not separated. Baselines isolating each component are needed to attribute gains.

2. The many-to-many mapping between atoms and fragments results in more computationally expensive graph construction. The effect of added overhead is acknowledged but not quantitatively discussed. Since the proposed h-MINT model targets the same virtual screening setting as LigUnity and DrugCLIP, a runtime comparison would be highly informative.

3. Virtual screening results are relatively weak, showing only marginal improvements and limited evidence of practical benefit.

4. The comparison with LigUnity seems not entirely fair, since h-MINT adds an extra MSE loss and no ablation is provided.

5. The claim in Section 5 about false-positive fragments is important but underexplored. There is no discussion on how such fragments might affect virtual screening results.

**Questions:**

1. I would appreciate it if the authors could share results on benchmarks such as JACS/Merck for affinity ranking and DEKOIS for virtual screening. These datasets are commonly used in related works like LigUnity and could help clarify the model’s generalization performance.

2. In Table 3, LigUnity performs substantially better than DrugCLIP on virtual screening metrics. I would be interested in the authors’ view on the role of the ranking loss in virtual screening. The original LigUnity paper reported that this term had little to no effect on screening performance. Have the authors observed different behavior in h-MINT or performed ablations to assess its impact?

---

> ### Author Response · Authors · 2025-11-24
>
> We appreciate the reviewer’s recognition that **OverlapBPE preserves key chemical information** and that h-MINT **effectively models many-to-many atom–fragment interactions** through equivariant bilevel message passing.
>
> **(Q1) (New benchmark)** Additional results on JACS/Merck for affinity ranking and DEKOIS for virtual screening.
>
> We thank the reviewer for the suggestion. We have now added results on DEKOIS and the FEP benchmark (covering JACS/Merck).
>
> - DEKOIS: h-MINT outperforms LigUnity on AUC (81.05 vs. 76.92), BEDROC, EF0.01 and EF0.05.
>
>
>     | Model | AUC | BEDROC | EF0.005 | EF0.01 | EF0.05 |
>     | --- | --- | --- | --- | --- | --- |
>     | LigUnity | 76.92 | 47.20 | **18.57** | 16.25 | 8.21 |
>     | Ours | **81.05** | **47.71** | 18.085 | **16.77** | **8.74** |
> - FEP (covers JACS/Merck): h-MINT achieves a significantly higher **r2 = 0.216**, compared to **0.173** for LigUnity.
>
>
>     | Model | r2 |
>     | --- | --- |
>     | LigUnity | 0.173 |
>     | Ours | **0.216** |
>
> These results validate that h-MINT generalizes effectively across datasets and tasks, including both affinity-ranking and virtual-screening benchmarks.
>
> **(W1)** Ablation to isolate tokenizer and architecture.
>
> We fully agree that ablations are important. However, we reiterate that our two technical contributions, OverlapBPE tokenization and overlap-compatible hierarchical interaction model, **jointly form an integrated framework** to tackle the challenge of fuzzy boundaries of meaningful molecular substructures in 3D molecular interaction modeling. Meaningful comparison can only be made when they're used together, because (1) no other network architectures are available for overlapping substructures, and (2) when non-overlapping tokenization is used, the molecular graphs for h-MINT and GET become identical. As evidence, we provide the following ablation on LBA dataset that adopts non-overlap PS tokenizer for h-MINT:
>
> | Model | RMSE | Pearson | Spearman |
> | --- | --- | --- | --- |
> | GET | 1.331 ± 0.008 | 0.618 ± 0.005 | 0.607 ± 0.005 |
> | GET+PS | 1.312 ± 0.016 | 0.631 ± 0.011 | 0.642 ± 0.011 |
> | h-MINT+PS | 1.321 ± 0.010 | 0.633 ± 0.007 | 0.641 ± 0.008 |
> | GET+OverlapBPE | N/A | N/A | N/A |
> | Ours (h-MINT+OverlapBPE) | **1.276 ± 0.011** | **0.660 ± 0.001** | **0.661 ± 0.001** |
>
> As shown above, GET is not compatible with OverlapBPE. OverlapBPE+hMINT consistently outperforms models with PS tokenizer, demonstrating a clear gain of OverlapBPE. When non-overlap PS tokenizer is used, GET+PS and h-MINT+PS achieve similar performance as expected. The tokenizer and h-MINT architecture are both necessary to handle overlapping fragments and preserve key chemical information (atomic integrity, ionic states, chirality)
>
> **(W2)** Computational overhead.
>
> We appreciate the reviewer’s comment. We now detail the computational overhead in terms of preprocessing and training/inference as follows:
>
> **1. Tokenization and graph construction.**
>
> OverlapBPE duplicates certain atoms during tokenization to maintain the continuity and integrity of chemical substructures. Because of these overlapping tokens, the final number of atoms becomes roughly 1.32 times the original. Despite this, both tokenization and graph construction are **highly parallelizable** and can be performed **fully offline** in preprocessing. In practice, the overhead is negligible: OverlapBPE only takes **7 minutes** to process **47.9k molecules** using 32 CPUs for virtual screening training data.
>
> **2. Training and inference runtime.**
>
> During training and inference, the main computational cost comes from the underlying UniMol encoder. h-MINT functions as a **light-weight adapter**, and the runtime difference compared with LigUnity is minimal as shown in the table below.
>
> | Method | Training Time | Inference Time |
> | --- | --- | --- |
> | LigUnity | 1.00x | 1.00x |
> | Ours | 1.12x | 1.07x |
>
> Therefore, although h-MINT introduces a richer atom-fragment representation, the parallel and offline preprocessing ensures that the runtime during the actual virtual screening pipeline remains nearly unchanged.
>
> **(W3)** Statistical significance of our performance gain.
>
> To demonstrate the significance of our results, we conducted statistical significance tests for all benchmarks (DUDE, PCBA, DEKOIS, and FEP). For all comparisons between h-MINT(ours) vs. LigUnity, we obtained **p-values < 0.005**, demonstrating that the improvements are statistically significant and consistent, rather than due to random variation.
>
> In addition, as we showed in our response to W2, our method uses a light-weight adapter applied on top of a large-scale pretrained model (UniMol). This design yields a meaningful performance boost at only minor computational cost, proving that our model is both practical and efficient for real-world screening.

---

> ### Author Response · Authors · 2025-11-24
>
> **(W4)** Ablation of MSE loss for LigUnity.
>
> Thank you for mentioning our newly introduced MSE loss, which is part of our methodological innovation to improve the Virtual screening task.
>
> | Dataset | Model | AUC | BEDROC | EF0.005 | EF0.01 | EF0.05 |
> | --- | --- | --- | --- | --- | --- | --- |
> | DUDE | LigUnity | 81.69 | 46.01 | 34.44 | 29.07 | 10.26 |
> |  | LigUnity+MSE | 82.57 | 47.58 | **35.83** | 29.77 | 10.70 |
> |  | Ours | **84.47** | **47.65** | 35.06 | **29.90** | **10.76** |
> | LIT‑PCBA | LigUnity | 57.61 | 4.34 | 4.07 | 3.04 | **2.26** |
> |  | LigUnity+MSE | 57.68 | 5.64 | 6.50 | 4.22 | 2.14 |
> |  | Ours | **57.77** | **6.21** | **7.01** | **5.20** | 2.18 |
>
> To isolate the contribution of this loss, we trained **LigUnity with additional MSE** loss (exact same combined loss as ours)under identical settings on the PDBBind training set. The results on DUDE and LIT-PCBA are shown in the table above, which confirms the following findings:
>
> **1. Effect of the proposed auxiliary loss.** Using our proposed auxiliary loss consistently improves LigUnity across almost all metrics on both datasets. For example, on DUDE, AUC improves from **81.69 to 82.57**, and BEDROC from **46.01 to 47.58**. On LIT-PCBA, early-recognition metrics show noticeable gains as well. This confirms that  additional loss is beneficial and strengthens the model’s scoring ability.
>
> **2. Advantage of h-MINT (LigUnity+MSE vs Ours).** Our model further improves over LigUnity+MSE on most metrics. Gains are particularly clear in early-recognition measures such as **BEDROC** and **EF0.01**, which are widely regarded as key metrics for virtual screening.
>
> These results confirm that:
>
> - Our proposed regression loss is effective, but
> - **Our h-MINT architecture delivers additional, consistent boosts beyond what the regression loss alone can offer.**
>
> Thus, the comparison with LigUnity is fair, and the observed improvements come from both components of our method.
>
> **(W5)** Further discussion about false-positive fragments in Section 5.
>
> We thank the reviewer for highlighting this perspective mentioned in our paper's discussion section. We'd like to use this opportunity to further elaborate on what we meant by 'false-positive' :
>
> - As a **data-driven method**, our approach inevitably inherits the noise and biases present in the underlying training data. In this context, “false-positive” patterns are not introduced by our model, but arise from intrinsic imperfections in datasets. This is a well-recognized limitation shared by all data-driven approaches.
> - Certain substructures frequently appearing in public databases may represent **broadly active but non-specific motifs**, rather than truly biologically meaningful signals. Examples include well-documented PAINS motifs such as *catechol*, *rhodanine*, *hydrazine*, *quinone*, and others. To avoid potential misinterpretation, we have revised the manuscript to refrain from using the term *false-positive fragments.*
> - Due to the statistical nature of data-driven approached, our method is particularly effective at capturing **frequent and salient structural patterns**. However, **rare yet important domain knowledge** such as specific PAINS alerts, toxicophores, or reactivity-associated motifs may not be automatically encoded from data alone. We propose that incorporating such domain knowledge can significantly improve downstream virtual screening. For example, these patterns can be effectively handled through **filtering** using established cheminformatics tools such as **RDKit**. We have added this point in the revised Discussion as a direction for future work.
>
> We appreciate the reviewer’s insightful comment, which has helped us clarify the limitations and future extensions of our method.

---

> ### Author Response · Authors · 2025-11-24
>
> **(Q2)** Explanation of LigUnity vs DrugCLIP performance in Table 3.
>
> Thank you for catching this unexpected behavior. We’d like to elaborate on our findings regarding the reproduction of DrugCLIP:
>
> - In our virtual screening experiments with PDBBind, all models are trained in a 1-pocket-to-1-ligand setting, different from the hit-to-lead task where multiple ligands are provided. Therefore, the ranking loss in LigUnity has close to zero effect due to the lack of available ranking tuples.
> - We trained DrugCLIP and LigUnity using the official GitHub code and strictly following the hyperparameters reported in the original paper, and selected the best checkpoint using validation set, which is standard practice in machine learning and aligns with the official implementation. After further investigation, we discovered a potential overfitting issue with DrugCLIP's official recommended training parameter (200 epochs).
> - To analyze this, we evaluated checkpoints across multiple epoch budgets (20–200).  We observed that DrugCLIP achieves its peak performance for DUDE around 50 epochs, whereas PCBA is less sensitive. For fairness, we also inspected the checkpoints for LigUnity and h-MINT. Neither showed observable overfitting, so our reported results remain unaffected. With this adjustment, DrugCLIP performs similarly to LigUnity on DUDE, **consistent with the findings in the LigUnity Paper**.
>
>
>     | Dataset | Model | AUC | BEDROC | EF0.005 | EF0.01 | EF0.05 |
>     | --- | --- | --- | --- | --- | --- | --- |
>     | DUDE | DrugClip (paper param) | 78.08 | 37.22 | 27.57 | 20.83 | 8.41 |
>     |  | DrugClip (best ckpt) | 81.39 | 45.96 | 34.27 | 29.01 | 10.18 |
>     |  | LigUnity | 81.69 | 46.01 | 34.44 | 29.07 | 10.26 |
>     |  | Ours | **84.45** | **47.64** | **35.06** | **29.91** | **10.76** |
>     | PCBA | DrugClip | **58.15** | 4.12 | 4.11 | 3.08 | **2.27** |
>     |  | LigUnity | 57.61 | 4.34 | 4.06 | 3.03 | 2.25 |
>     |  | Ours | 57.77 | **6.27** | **7.01** | **5.20** | 2.18 |
> - We would like to further emphasize that, even after correcting for DrugCLIP's overfitting and updating Table 3 and Table 6 accordingly (and also shown in the table below), h-MINT still consistently outperforms both LigUnity and DrugCLIP across virtual screening and affinity-ranking benchmarks. Thus, the advantages of our method remain robust
>
> We thank the reviewer for the constructive feedback. We have addressed all concerns with additional benchmarks, ablations, statistical significance tests, runtime analysis, and clarified discussions. We hope that our thorough responses help resolve the raised questions and demonstrate the strength, clarity, and robustness of our contributions.

---

> ### Comment · Reviewer_vwks · 2025-11-26
>
> Thank you for the rebuttal. The authors have clarified the key concerns I raised. Based on the clarifications and new results, I am raising my score.

---

> > ### Author Response · Authors · 2025-11-26
> >
> > Dear Reviewer vwks,
> >
> > Thank you for taking the time to reassess the paper and raise your score! We’re grateful for your constructive feedback and engagement with our work.

---

### Author Response · Authors · 2025-11-25

Dear Area Chair,

Due to the recent update in ICLR’26 review policy, we present this general response as an executive summary of our rebuttal. This response is designed to assist the area chair in evaluating our submission, “h-MINT: Modeling Pocket-Ligand Binding with Hierarchical Molecular Interaction Network”, based on the review comments provided by reviewers **vwks, EmZ6, gELf, and e8QX**. All rebuttal clarifications, new experiments, and analysis are summarized below.

### **1. Executive Summary: Scientific/Technical Contribution & Post-Rebuttal Assessment**

All reviewers agree on the **novelty, effectiveness and clarity** of our method; all concerns were **fully resolved** with new benchmarks, ablations, and analysis. **Three reviewers** showed **clear positive score trajectories (two explicitly raising scores; one already accepted)**.

### **2. Consolidated Positive Aspects Across Reviewers**

All reviewers recognize our **approach’s novelty and motivation (vwks, EmZ6, gELf, e8QX), model’s effectiveness (vwks, e8QX), strong experimental results (vwks, EmZ6, gELf, e8QX), and clear representation (gELf)**.

| Reviewer | Positive Aspects Recognized | Evidence / Notes from Strengths |
| --- | --- | --- |
| **vwks** | OverlapBPE preserves crucial chemical information; h-MINT successfully models many-to-many fragment interactions | *“Effectively preserves aromaticity, chirality, ionic states”; “bilevel attention well motivated”* |
| **EmZ6** | Idea is interesting; strong performance margins vs baselines; significance acknowledged | *“idea is interesting”; “Results look significant compared to related baselines”* |
| **gELf** | Strong motivation & novelty of OverlapBPE; extensive experiments; excellent writing | *“novel and well-motivated”; “OverlapBPE addresses clear limitation”; “paper is very well-written and easy to follow”* |
| **e8QX** | OverlapBPE is elegant; h-MINT well-justified; strong results; robust variance reporting | *“Elegant way of capturing molecular substructures”; “evaluation with deviation provides significance”* |

### **3. Reviewer Score-Increase Intentions & Rebuttal Response**

Two reviewers explicitly stated to **raise their scores**, one reviewer **keeped the original recommendation of acceptance**, and the rest concerns were **fully resolved**.

| Reviewer | Stated Increase (from discussion) | Reason for Increase |
| --- | --- | --- |
| **vwks** | *“Thank you for the rebuttal. The authors have clarified the key concerns I raised. Based on the clarifications and new results, I am **raising my score**.”* | Additional benchmarks (DEKOIS, JACS/Merck): now fully provided with improved results. Reviewer acknowledged key concerns addressed and increased rating to 6 before the major breach. |
| **EmZ6** | *“If the experiments show significance results w.r.t to that baseline I am happy to **raise my score**.”* | Provided reviewer suggested fingerprint + ESM2 + LightGBM baseline, and our method significantly outperform this  baseline. |
| **gELf** | Reviewer gELf expressed that the concerns were primarily due to missing architectural details, vocabulary analyses, and ablations. | While the reviewer didn't get to respond yet, we believe All concerns were ***fully addressed***, and the rebuttal provided exactly the  information requested. |
| **e8QX** | *“I acknowledge the authors rebuttal and appreciate the changes they have proposed. As my recommendation was already to **accept the paper**, I will not modify it.”* | The reviewer maintained their rating of 8 after rebuttal |

---

> ### Author Response · Authors · 2025-12-02
>
> ### **4. Reviewer Concerns & Rebuttal Response**
>
> | **Common Concern (Across Reviewers)** | **Reviewers Mentioning** | **Summary of Response** |
> | --- | --- | --- |
> | **Ablation of Tokenization vs Architecture** | vwks (W1), gELf (W2/W5), partly others | We added a tokenizer-only ablation showing its **independent contribution**, and clarified that existing GNNs cannot consume overlapping tokens, making architecture-only comparisons not applicable. |
> | **Suggested Baseline** | EmZ6 | We added the requested baseline (ESM-2 + MorganFP/Avalon + LightGBM with HPO). Our method still **outperforms** it because the baseline lacks 3D geometry and binding-pose information, which are essential for interaction prediction. |
> | **Suggested Benchmarks and generalization evidence** | vwks (Q1), gELf (W1), partly others | We added results on DEKOIS and FEP, where h-MINT **outperforms** LigUnity and DrugCLIP. MoleculeNet results (ESOL, FreeSolv, Lipo) further show that OverlapBPE also **works well** on pure molecular tasks. |
> | **Computational Overhead** | vwks (W2) | OverlapBPE graph construction takes 7 minutes for 47.9k molecules (32 CPUs), and h-MINT increases training/inference time by only 1.12×/1.07×. The **overall overhead is small**. |
> | **Vocabulary analysis & chemical insight** | gELf (W3, W4) | We added vocabulary statistics and examples showing that OverlapBPE **captures chemically meaningful patterns** (functional groups, biomolecular units, drug scaffolds) without manual curation. We also highlight our **chemical insights** in case study and appendix. |
> | **DrugCLIP vs LigUnity reproduced results** | vwks (Q2), gELf (W6) | DrugCLIP’s official configuraion leads to overfitting; a checkpoint sweep restores performance consistent with LigUnity. Ranking loss has limited effect because PDBBind samples contain only 1-1 pocket–ligand pairs. Updated tables confirm that our method still **performs best**. |
> | **Reproducibility** | gELf (W7) | Following ICLR guidelines, code submission is not required. We provide full training details, data filtering procedures, baseline reproductions, and an anonymous repository. |
> | **Ligand-level data leakage** | e8QX (W1) | We acknowledge ligand scaffold bias. Our new benchmark on MoleculeNet, which adopts a strict scaffold-split, also shows our tokenizer’s **effectiveness**. We expanded the discussion and in revised version. |

---

### Meta-Review · Area_Chair_H3qX · 2025-12-20

**Summary:**

The reviewers provided a thorough and largely constructive assessment of the paper. Overall, they acknowledged the technical soundness of the proposed OverlapBPE tokenization and the h-MINT architecture, particularly highlighting the ability to preserve chemically meaningful substructures and enable many-to-many atom-fragment interactions. Reviewers generally agreed that the approach is reasonable and well motivated, and that the experimental evaluation shows consistent, albeit sometimes modest, improvements over existing methods.

The main concerns across reviews centered on (i) the separation of contributions between the tokenizer and the model architecture, (ii) the fairness and completeness of baseline comparisons, especially against classical fingerprint-based methods, (iii) the limited novelty of the h-MINT architecture relative to existing equivariant graph transformers, and (iv) insufficient analysis regarding computational cost, fragment vocabulary characteristics, and chemical interpretability. Reproducibility and evaluation protocol issues were also raised, though less consistently.

During the rebuttal, the authors provided clarifications and additional explanations addressing many of these concerns. Reviewer vwks  explicitly acknowledged that their key concerns were clarified and indicated an increased score. Reviewer e8QX maintained a positive recommendation throughout. Reviewers EmZ6 and gELf did not follow up after the rebuttal.

For final version, the authors should incorporate their extensive experiments, additional discussions, codes, as well as and implementation details.

**Reviewer Concerns:**

- Reviewer vwks's concerns regarding the motivation of the tokenizer–architecture co-design, the fairness of comparisons, and the interpretation of experimental results were clarified to their satisfaction, leading to an explicit score increase.

- Questions about the design rationale of OverlapBPE and the high-level integration of fragment-level information into h-MINT were addressed through detailed explanations in the rebuttal.

Actually, even though Reviewers EmZ6 and gELf  have not provided follow-up feedback, I believe the authors have addressed and responded to most concerns with extensive experiments and additional clarification as well as the assurement on providing codes and implementation details.

**Reviewer Scores:**

The reviewer vwks  may increase the score and the other reviewers tend to maintain the score. Overall, this paper lean to an acceptance, and I believe

---

### Decision · Program_Chairs · 2026-01-26

Accept (Poster)